# Repression of interrupted and intact rDNA by the SUMO pathway in *Drosophila melanogaster*

Yicheng Luo[1], Elena Fefelova[1,2], Maria Ninova[1], Yung-Chia Ariel Chen[1], Alexei A Aravin[1]*

[1]Division of Biology and Biological Engineering, California Institute of Technology, Pasadena, United States; [2]Institute of Molecular Genetics, Russian Academy of Sciences, Moscow, Russian Federation

**Abstract** Ribosomal RNAs (rRNAs) are essential components of the ribosome and are among the most abundant macromolecules in the cell. To ensure high rRNA level, eukaryotic genomes contain dozens to hundreds of rDNA genes, however, only a fraction of the rRNA genes seems to be active, while others are transcriptionally silent. We found that individual rDNA genes have high level of cell-to-cell heterogeneity in their expression in *Drosophila melanogaster*. Insertion of heterologous sequences into rDNA leads to repression associated with reduced expression in individual cells and decreased number of cells expressing rDNA with insertions. We found that SUMO (Small Ubiquitin-like Modifier) and SUMO ligase Ubc9 are required for efficient repression of interrupted rDNA units and variable expression of intact rDNA. Disruption of the SUMO pathway abolishes discrimination of interrupted and intact rDNAs and removes cell-to-cell heterogeneity leading to uniformly high expression of individual rDNA in single cells. Our results suggest that the SUMO pathway is responsible for both repression of interrupted units and control of intact rDNA expression.

**\*For correspondence:**
aaa@caltech.edu

**Competing interests:** The authors declare that no competing interests exist.

## Introduction

Ribosomal RNAs (rRNAs) are the main structural and central enzymatic components of ribosomes. The constant production of ribosomal RNA is essential for cell growth and division. The high demand for rRNA production is addressed by a two-prong strategy. First, cells use a dedicated transcription machinery composed of RNA polymerase I and associated unique transcription factors to transcribe rDNA genes. Second, genomes contain multiple identical rDNA genes that are transcribed simultaneously.

In most eukaryotes rRNA genes encoding the 18S, 28S, and 5.8S rRNAs are transcribed under the control of a single promoter producing pre-rRNA (*Mandal, 1984*). After co-transcriptional processing and modification, the pre-RNA is converted to mature 18S, 28S, and 5.8S rRNAs, which together with the additional 5S rRNA and ribosomal proteins are assembled into pre-ribosomal subunits (*Henras et al., 2015*). Actively transcribed rDNA genes and pre-ribosomal subunits at different steps of maturity form the nucleolus – a membraneless compartment in the nucleus characterized by tripartite morphological ultrastructure.

In eukaryotic genomes dozens to hundreds of nearly identical rDNA units are arranged head-to-tail forming one or several clusters. Studies in such diverse organisms as different plants, yeast, fruit flies, mice and humans showed that only fraction of available rDNA genes is transcriptionally active (*Bird et al., 1981*; *Coffman et al., 2005*; *Conconi et al., 1989*; *Dammann et al., 1993*; *Flavell et al., 1988*). In organisms that have several rDNA clusters located on different chromosomes, whole clusters might be transcriptionally inactive and positioned outside of the nucleolus.

For example, in Drosophila where rDNA clusters are located on the X and Y chromosomes, in certain genotypes only the Y chromosome cluster is active and forms the nucleolus (*Greil and Ahmad, 2012*; *Zhou et al., 2012*). In mammals which carry several ribosomal loci on different chromosomes some clusters are constantly active, while the activity of others depends on the tissue type, presence of nutrients and stress (*Ali et al., 2008*; *Ali et al., 2012*; *de Capoa et al., 1985a*; *de Capoa et al., 1985b*; *Tseng et al., 2008*; *Young et al., 2007*; *Zhang et al., 2007*). Direct observation of rDNA transcription by electron microscopy showed that individual rDNA units within a single cluster might also have different transcriptional activity and are subject to all-or-none regulation: any given rDNA unit is either actively transcribed with multiple RNA pol I associated with nascent pre-rRNA positioned along the body of the unit, or is inactive. (*Foe, 1978*; *McKnight and Miller, 1976*).

Due to recombination-driven homogenization, individual rDNA units are similar in sequence (*Eickbush and Eickbush, 2007*), arguing against difference in sequence leading to differential expression. In fact, repressed rDNA units might become active when the genetic environment or demand for rRNA production changes. For example, in Drosophila X-linked rDNA genes that are inactive in males are active in females (*Zhou et al., 2012*). The molecular mechanism for differential activity of rDNA units was studied in several organisms. In mammals repressed rDNA units are enriched in DNA methylation, repressive histone marks (*Coffman et al., 2005*; *Earley et al., 2006*; *Li et al., 2006*; *Santoro and Grummt, 2001*; *Santoro et al., 2002*; *Zhou et al., 2002*). The chromatin remodeling complexes NoRC and NuRD, in cooperation with noncoding RNAs derived from rDNA loci were shown to be involved in rDNA repression (*Santoro et al., 2002*; *Strohner et al., 2001*; *Xie et al., 2012*; *Zhou et al., 2002*; *Bierhoff et al., 2014*; *Mayer et al., 2006*; *Santoro et al., 2010*). However, a deep understanding of the molecular mechanisms establishing and maintaining rDNA repression is lacking.

Genomes of many arthropod species including *Drosophila melanogaster* harbor transposable elements that integrate into rDNA, creating distinct rDNA units. R1 and R2 belong to the non-long terminal repeat (non-LTR) retrotransposons and encode a sequence-specific endonuclease that is responsible for integration of these elements into 28S rDNA (*Burke et al., 1987*; *Eickbush and Eickbush, 2015*; *Jakubczak et al., 1990*; *Xiong et al., 1988*; *Yang et al., 1999*). It is believed that R1 and R2 lack their own promoters and instead rely on the rDNA transcription machinery for their expression by transcribing as part of pre-rRNA followed by excision mediated by the transposon-encoded ribozyme (*Eickbush and Eickbush, 2010*; *Eickbush et al., 2013*).

The fraction of rDNA units with R1 and R2 insertions varies between different *D. melanogaster* strains, but was estimated to reach up to 80% of units in some strains (*Jakubczak et al., 1992*). Despite the abundance of R1 and R2, their expression level is usually low. Indeed, electron microscopy and run-on analysis of nascent transcripts revealed that rDNA units with transposon insertions are transcriptionally silent, indicating a mechanism that can distinguish interrupted rDNA copies and repress their transcription (*Jamrich and Miller, 1984*; *Ye and Eickbush, 2006*). Interestingly, R1 and R2 expression is increased if the total number of rDNA units in the genome is low, suggesting that repression is sensitive to cellular rRNA demand (*Eickbush and Eickbush, 2003*; *Long et al., 1981*; *Terracol, 1986*). The molecular mechanism for repression of interrupted rDNA units and its relationship to the general mechanism of rDNA silencing remained unknown.

SUMO (Small Ubiquitin-like Modifier) is a small protein related to ubiquitin that is covalently attached to other proteins in sequential reactions mediated by E1, E2 and, for some targets, E3 SUMO ligases. Unlike ubiquitination, SUMOylation typically is not linked to protein degradation and instead changes protein-protein interactions by facilitating recruitment of new binding partners or masking existing binding sites. The majority of SUMOylated proteins reside in the nucleus and SUMOylation was shown to be involved in regulation of transcription, mRNA processing, chromatin organization, replication and repair (*Geiss-Friedlander and Melchior, 2007*) as well as rRNA maturation and nucleolus function (*Finkbeiner et al., 2011*; *Haindl et al., 2008*; *Yun et al., 2008*; *Westman et al., 2010*). SUMOylation was shown to be essential for growth and viability of *Saccharomyces cerevisiae* (*Johnson et al., 1997*), *D. melanogaster* (*Lehembre et al., 2000*) and mice (*Nacerddine et al., 2005*). Mammalian genomes encode four distinct SUMO genes with partially redundant functions. In contrast, only one SUMO gene is present in the *D. melanogaster* genome, making it a good model to understand the diverse functions of SUMOylation (*Guo et al., 2004*; *Melchior, 2000*; *Yang et al., 1999*).

Here we have established a new model to study regulation of rDNA expression using a molecularly marked single-unit rDNA transgene. We found that this model faithfully recapitulates repression of interrupted rDNA units. We discovered that repression of rDNA units interrupted by transposon insertions, as well as intact rDNA units is controlled by the SUMO pathway, indicating that the same molecular mechanism is responsible for epigenetic inactivation of intact and interrupted rDNA.

## Results

### A single-unit rDNA transgene is expressed from a heterologous genomic site

The *D. melanogaster* genome contain numerous (50-200) rDNA units arranged in tandem repeats in the heterochromatin of the X and Y chromosomes (*Figure 1A*). Many rDNA units contain insertions of R1 and R2 transposons that lack their own promoters and co-transcribe with pre-rRNA (*Ye and Eickbush, 2006*). However, expression of R1 and R2 elements is low in the majority of *Drosophila* strains, indicating that units with insertions are specifically repressed (*Ye and Eickbush, 2006*). The large number of identical rDNA units organized in large arrays makes it impossible to study expression of individual rDNA units by conventional methods. To circumvent this problem and study regulation of rDNA expression, we created flies with a transgene that carries a molecularly marked single rDNA unit inserted into a heterologous genomic locus (*Figure 1A*). It harbors the non-transcribed spacer (NTS), which includes elements that regulate RNA polymerase I (Pol I) transcription and the complete transcribed portion, which generates the 47S pre-rRNA. The pre-rRNA contains a 5' external transcribed sequence (5' ETS) and internal transcribed spacers (ITS), which are removed from the pre-rRNA during its processing into mature 18S, 5.8S, 2S and 28S rRNAs. To discriminate transcripts generated from the transgene from endogenous rRNA we inserted a 21 bp unique identification sequence (UID) into the external transcribed spacer (ETS) downstream of the transcription start site (*Figure 1A*). Along with the rest of the ETS, the UID sequence is removed from pre-rRNA during its processing to mature rRNAs in the nucleolus, however, it allowed us to monitor transgene transcription. In contrast to endogenous rDNA arrays, which are located in heterochromatin of the X and Y chromosomes, the single-unit rDNA transgene was integrated into a euchromatic site on 2nd chromosome (chr 2L: 1,582,820) using site-specific integration.

First, we tested if the rDNA transgene is expressed in the heterologous location. RT-qPCR with one primer specific to UID sequence detected expression of the rDNA transgene in transgenic flies, while no PCR product was produced in the parental strain used for transgenesis (*Figure 1B*). Thus, RT-qPCR is able to discriminate expression of the marked rDNA transgene from endogenous rDNA and insertion of UID sequence does not disrupt the Pol I promoter and enhancer elements.

Next, we employed fluorescent in situ hybridization (FISH) to detect expression of the rDNA transgene in individual cells. Nascent transgene transcripts differ from more abundant endogenous pre-rRNA by a 21 nt UID sequence, however, the standard FISH protocol using short probe against UID failed to detect expression of the transgene. To circumvent low sensitivity of standard FISH, we used a method based on the mechanism of the hybridization chain reaction (HCR-FISH) that offers a combination of high sensitivity and quantitation (*Choi et al., 2018*). HCR-FISH allowed specific detection of nascent transgene transcripts and revealed that transgene RNA is present exclusively in the nucleus (*Figure 1C*) consistent with the ETS portion being processed out of pre-rRNA and degraded soon after transcription. Thus, the marked rDNA transgene enables expression analysis of an individual rDNA unit inserted in a heterologous genomic location.

### Insertion of a heterologous sequences into rDNA leads to decreased expression

In *Drosophila* significant fraction of rDNA units contain insertions of R1 and R2 retrotransposons, however, units interrupted by transposon insertions are usually silent (*Long and Dawid, 1979*; *Jamrich and Miller, 1984*). R1 and R2 each integrate into single unique site positioned in 28S RNA at 2711 nt and 2648 nt of 28S (*Jakubczak et al., 1990*), respectively. To understand how insertion of transposons into rDNA influences its expression, we generated a series of additional transgenes. The full-length R1 (5.3 Kb) and R2 (3.6 Kb) transposon sequences were inserted in their respective integration sites within the 28S rDNA (*Figure 2A*). We also generated transgenes that contain

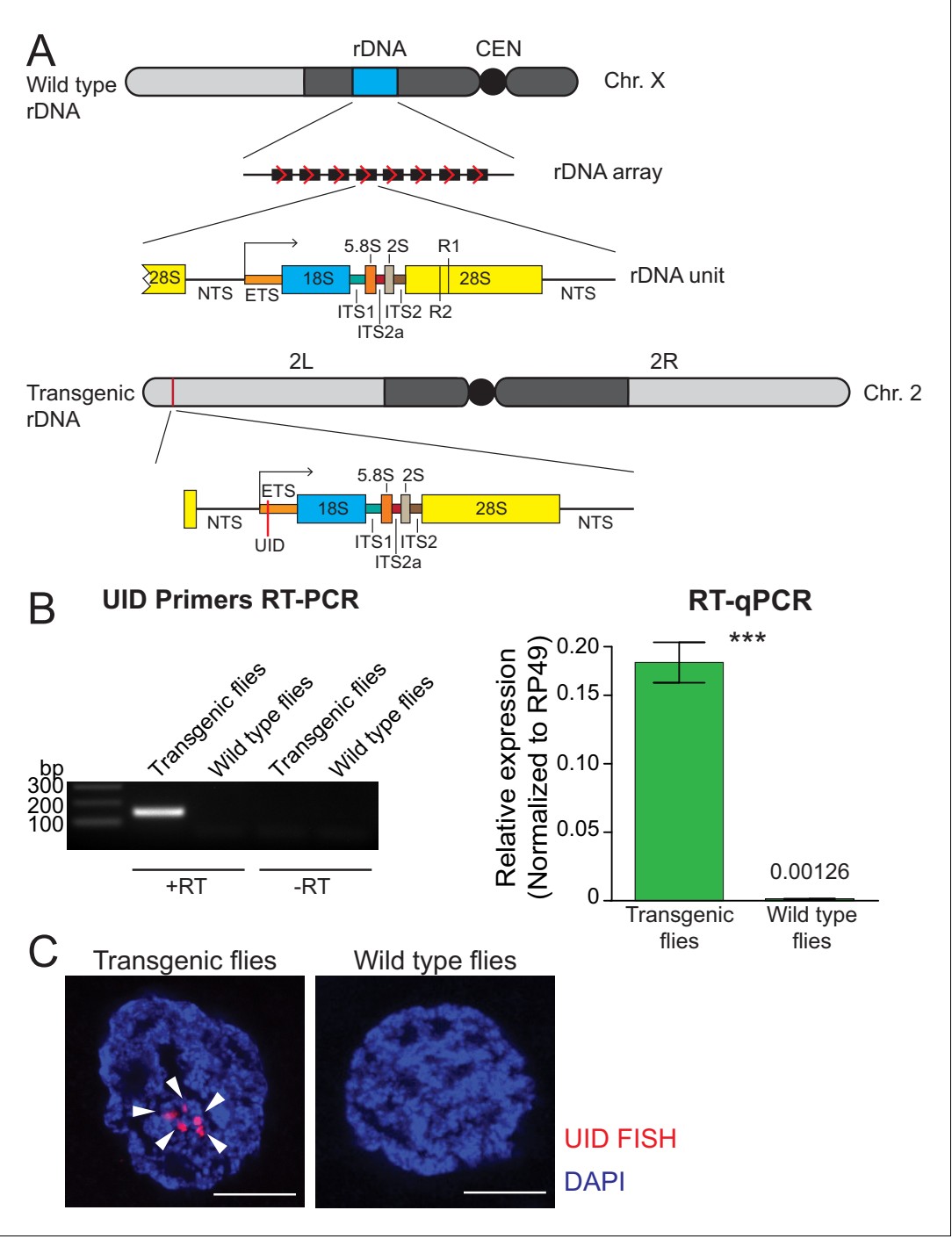

**Figure 1.** Single-copy rDNA transgene allows study of rDNA expression. (**A**) Scheme of endogenous rDNA units and the rDNA transgene. Native rDNA units are composed of non-transcribed spacer (NTS) and a transcribed portion that produces the 47S pre-rRNA. The pre-rRNA contains a 5' external transcribed sequence (ETS) and internal transcribed spacers (ITSs), which are removed from the pre-rRNA to generate mature 18S, 5.8S, 2S and 28S rRNAs. Positions of R1 and R2 transposon integration sites in the 28S rRNA are indicated. The 9 Kb transgene contains one complete rDNA unit together with a 28 bp of the 28S rDNA from the upstream unit. The transgene is marked by insertion of a 21 bp unique identification sequence (UID, red bar) into the ETS and inserted using ΦC31-mediated recombination into a common *att* site on the second chromosome (chr 2L: 1,582,820). (**B**) rDNA transgene expression is detected by RT-PCR in fly ovaries. RT-PCR amplicon of the UID ETS region is only detected in transgenic but not in wild-type flies or in the absence of reverse transcriptase (-RT). rDNA transgene expression in ovaries was measured by RT-qPCR and normalized to rp49 mRNA. Error bars indicate standard

*Figure 1 continued on next page*

*Figure 1 continued*

deviation of three biological replicas. Statistical significance is estimated by two-tailed Student's t-test; ***p<0.001. (C) rDNA transgene expression is detected by HCR-FISH in fly ovaries. Nascent transcripts of the pre-rRNA transgene (arrowhead) were detected in nurse cell nuclei using a probe against the UID sequence (red). Control wild-type flies lack the UID sequence. Scale bar: 5 µm.

insertion of an unrelated sequence (promoterless CFP) in the same positions as R1 and R2 (*Figure 2A*). In R1' and R2' transgenes, respective transposon sequences were flanked by additional UIDs. All transgenes were integrated into the same genomic location, which allows us to directly compare their expression.

Expression analysis of transgenes by RT-qPCR in ovary and carcass (*Figure 2B*) showed that transposon insertions lead to a 3- to 5-fold reduction in pre-rRNA level compared to the intact transgene. R1 insertion caused a slightly stronger reduction in expression compared to R2 insertion. Importantly, insertions of a non-transposon CFP sequence lead to comparable decrease in expression, indicating that the repressive effect is not dependent on a specific transposon sequence and can be triggered by heterologous sequence. A 29 bp insertion caused a 2-fold decrease in transgene expression, demonstrating that short insertions can affect rDNA expression, though to a lesser degree. Thus, expression of the single-copy rDNA transgene recapitulates the behavior of endogenous rDNA and demonstrates repression of units interrupted by transposon insertions.

## Cell-to-cell variability in rDNA transgenes expression

To monitor expression of rDNA transgenes in individual cells we employed HCR-FISH. Nurse cells of the fly ovary are an ideal cell type to study transgene expression as each egg chamber is composed of 15 identical sister nurse cells with large nuclei. The chromosomes of nurse cells forming one egg chamber undergo several rounds of endoreplication. Due to chromosome polytenization the HCR-FISH signals of rDNA transgenes and the endogenous control gene, *vasa*, are localized to several distinct loci on chromatin that likely represent sites of nascent transcription (*Figure 2C*). *Vasa* gene was expressed at similar levels in 15 nurse cells forming egg chamber. Expression of endogenous pre-rRNA detected by FISH probe against ETS (which detects rRNAs coming from all rDNA copies) was also similar in the different nuclei. In contrast, the rDNA transgenes have variable levels of expression in individual nuclei within one egg chamber with some nuclei showing no signal at all, while others had variable signal intensity. Double FISH demonstrated that nascent transgenic transcripts often co-localize with endogenous pre-rRNA (*Figure 2C*) indicating that transgene locus is recruited to nucleolus.

To analyze the cell-to-cell heterogeneity we first measured the fraction of nurse cells in each chamber that expressed the transgene. The intact rDNA transgene is expressed in the majority (10 ± 1 out 15, 67%), but not all cells of the egg chamber (*Figure 2D*). The fraction of individual nuclei with detectable FISH signal drops to 27% for transgenes with R1 and R2 insertions (*Figure 2D*). Next, we measured the level of HCR-FISH signal intensity in individual nuclei that had detectable signal. The intact rDNA transgene showed high level of cell-to-cell variability in expression (*Figure 2E*). The signal intensity was ~30 fold lower for transgenes containing R1 and R2 insertions compared to intact rDNA transgene (*Figure 2E*). These experiments revealed an unexpected cell-to-cell variability in expression of rDNA transgenes that was not present for control protein-coding gene *vasa*. rDNA units with insertions are expressed in less than one-third of all nuclei and even the intact unit is not expressed in all cells. Thus, repression of rDNA transgenes with insertions differs from cell-to-cell: in the majority of cells repression is complete and in the remaining cells expression is strongly reduced.

## The SUMO pathway is required for repression of transposons integrated into native rDNA loci

In the *Drosophila* germline many transposon families are repressed by the piRNA pathway. We recently found that SUMO, encoded by the single *smt3* gene in *Drosophila* and the E3 SUMO-ligase Su(var)2–10/dPIAS are involved in piRNA-mediated transcriptional silencing of transposons in germline (*Ninova et al., 2020a*). SUMO is covalently attached to many nuclear proteins in a conserved pathway that consists of the E1 (Uba2/Aos1) and E2 (Ubc9) SUMO ligases (*Figure 3A*). SUMOylation

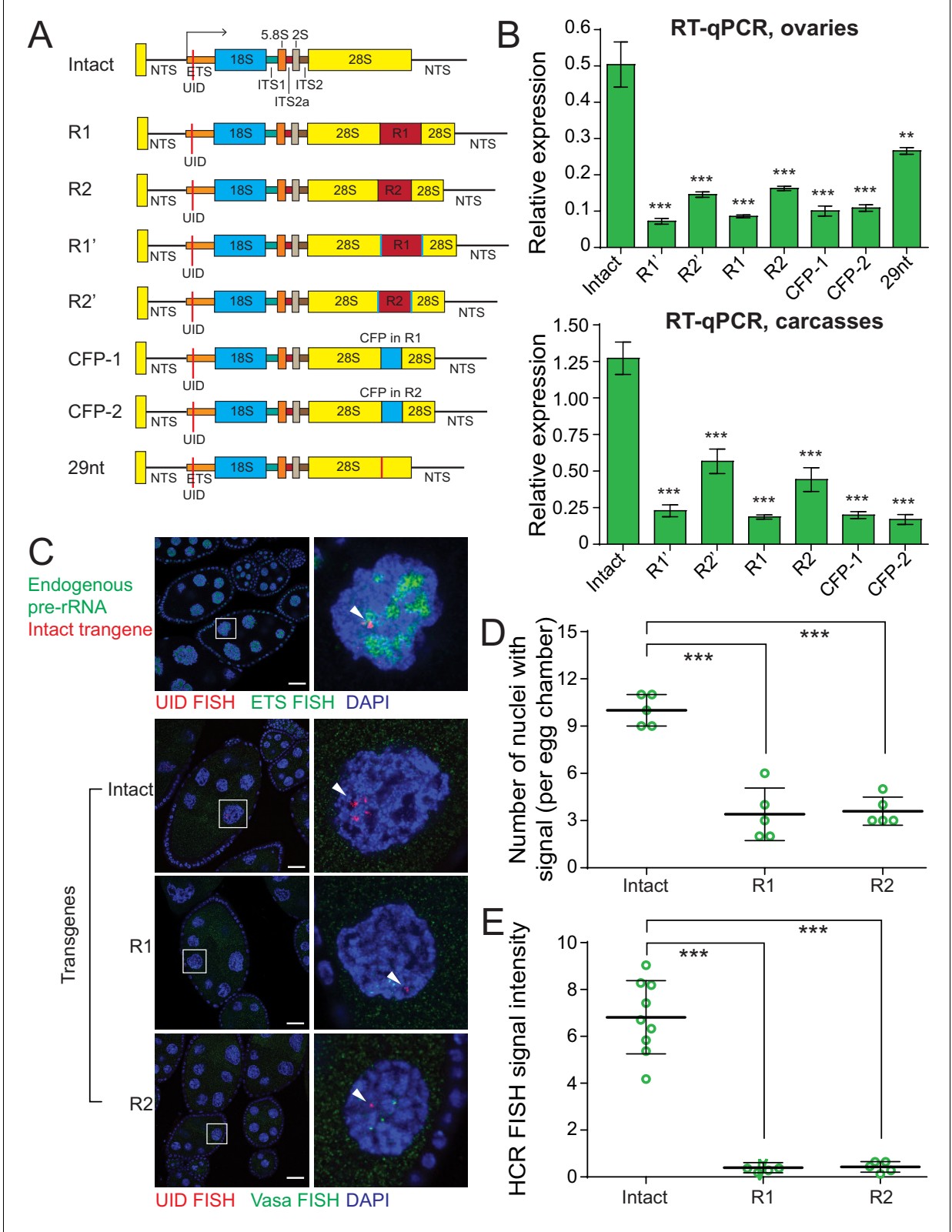

**Figure 2.** Insertions into 28S rRNA lead to decreased transgene expression. (**A**) The scheme of rDNA transgenes. The sequences of R1 and R2 transposons were inserted into their natural integration sites within the 28S rRNA. Transgenes R1' and R2' are identical to R1 and R2 transgenes, respectively, but contain second UID flanking retrotransposon sequence. The promotorless CFP sequences were inserted into the same R1 (CFP-1 transgene) and R2 (CFP-1 transgene) integration sites. 29nt sequence was inserted into R2 site. All constructs were integrated into the same genomic
*Figure 2 continued on next page*

*Figure 2 continued*

*att* site (chr 2L: 1,582,820) on the second chromosome using ΦC31-mediated recombination. (B) Transgenic rRNA expression is decreased upon insertion of foreign sequence. Expression of rDNA transgenes in ovary (top) and carcasses (bottom) measured by RT-qPCR and normalized to rp49 mRNA. Error bars indicate standard deviation of three biological replicas. Statistical significance is estimated by two-tailed Student's t-test; ***p<0.001. (C) R1 or R2 insertion decreases transgenic rRNA transcripts as detected by HCR-FISH Endogenous pre-rDNA is detected in stage 7–8 nurse cell nuclei using a probe against ETS (green), while transgenes are detected with probe against the UID (Red). Vasa mRNA is detected as a control (green). Scale bar: 10 μm. (D) Transgenic rDNA is expressed in fewer nuclei upon R1/R2 insertion Shown is the number of nurse cell nuclei with positive HCR-FISH signal per egg chamber (out of 15 nuclei per chamber). Each circle represents data from one egg chamber. Error bars indicate standard deviation from five egg chambers. Statistical significance is estimated by two-tailed Student's t-test; ***p<0.001. (E) Transgenic rDNA HCR-FISH signal is reduced upon R1/R2 insertion The total intensity of HCR-FISH signal was measured in individual nuclei that have positive signal. Error bars indicate standard deviation from five nuclei. Statistical significance is estimated by two-tailed Student's t-test; ***p<0.001.

of target proteins requires transfer of SUMO from Ubc9, which may occur directly or be aided by specific E3 SUMO ligases.

In agreement with the function of *smt3* and *Su(var)2–10* in piRNA-guided transposon repression, germline-specific knockdown (GLKD) of either gene causes similar activation of many TE families (*Ninova et al., 2020a*; *Ninova et al., 2020b*). Notably, GLKD of Su(var)2–10 had a modest effect on R1 and R2 compared to many other TE families that were upregulated more strongly. However, knockdown of *smt3* has a disproportionally strong effect on expression of R1 and R2 transposons. RNA-seq analysis showed that the levels of R1 and R2 transcripts increased ~1000 and~300 fold, respectively, and they become among the top 20 most abundant cellular mRNA (an average of 3164 and 3421 TPM (transcripts per million) in *smt3* GLKD, compared to 2.76 and 10.84 TPMs in control ovaries) (*Figure 3B,C*, *Figure 3—figure supplement 1*). The extremely strong R1 and R2 activation upon depletion of SUMO was also confirmed by RT-qPCR (*Figure 3D*). Analysis of RNA-seq data demonstrated that such strong up-regulation is exclusive for R1 and R2 and is not seen for any other protein-coding or ncRNA genes: the vast majority of other transposons and genes those expression was affected by *smt3* GLKD changed less than 10-fold. Thus, SUMO seems to be involved in specific process of R1/R2 repression.

To further explore the role of SUMO and piRNA in repression of R1 and R2 we knocked down *smt3* and the single E2 SUMO ligase, *Ubc9*, in S2 cells that lack an active piRNA pathway. RT-qPCR showed strong and specific upregulation of R1 and R2 upon both *smt3* and *Ubc9* knock-downs in S2 cells (*Figure 3E*). In contrast, knock-down of *Su(var)2–10* in S2 cells did not cause activation of R1 and R2 (*Figure 3—figure supplement 1*). Thus, SUMO is required for the potent and specific repression of R1 and R2 in both the soma and the germline, employing a mechanism that is independent of piRNA. R1 and R2 are embedded into rDNA and use the rRNA transcription machinery for their expression suggesting that SUMO controls expression of rDNA units interrupted by transposon insertions.

## SUMO depletion releases repression of interrupted and intact rDNA to equalize their expression

To test if SUMO regulates expression of the single-copy rDNA transgenes we studied their expression upon Smt3 knock-down by RT-qPCR. Transgenes with R1 and R2 insertions showed 54-fold and 50-fold increase in pre-rRNA expression level upon Smt3 KD (*Figure 4A*). Different transgenes with R1 and R2 as well as CFP insertions showed similar 41- to 54-fold increase in pre-rRNA level. Thus, the function of SUMO in repression of endogenous rDNA units with insertion of R1 and R2 transposons is recapitulated using rDNA transgenes. Surprisingly, expression of the intact rDNA transgene that lacks any insertion was also up-regulated ~12 fold upon SUMO KD (*Figure 4A*). While in wild-type flies interrupted rDNA copies are repressed compared to intact units (*Figure 2*), intact and interrupted copies have similar expression levels in the absence of SUMO. Thus, SUMO is required for differential expression of intact and interrupted rDNA. SUMO pathway participates in repression of both intact and interrupted rRNA, but does so with different efficiency.

To get further insights into expression of intact and interrupted rDNA transgenes upon SUMO depletion we employed HCR-FISH to study expression in individual nurse cells (*Figure 4B*). Confirming the RT-qPCR results, Smt3 GLKD lead to a marked increase in expression of both intact and interrupted rDNA transgenes. First, in contrast to wild-type flies, expression of intact and interrupted

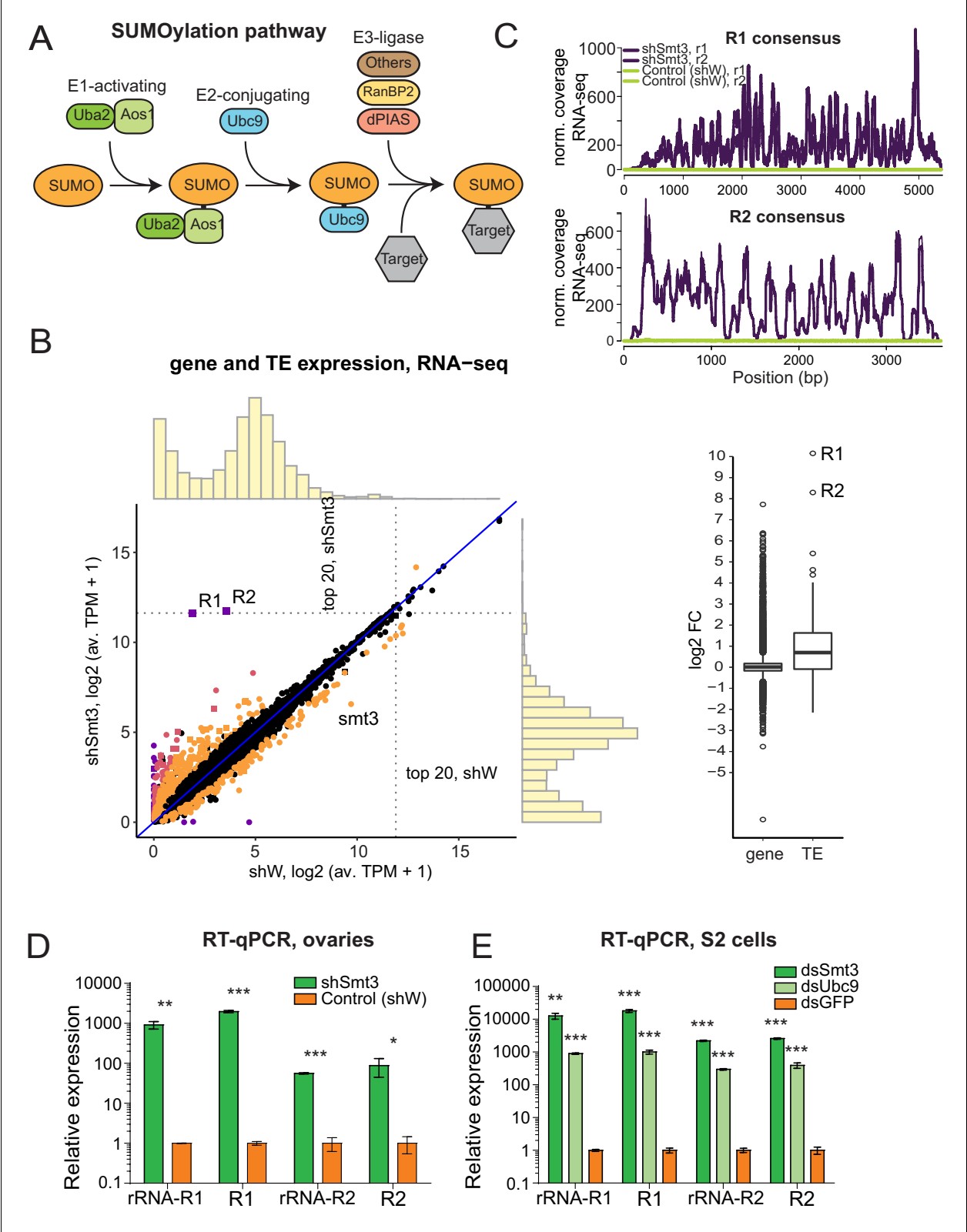

**Figure 3.** SUMO pathway is required for repression of transposons integrated into rDNA. (A) Scheme of SUMO pathway. SUMO is activated by E1 enzyme composed of Uba2/Aos1 dimer. Next SUMO is transferred from Aos1 to E2 Ligase Ubc9. Unc9 can directly transfer SUMO to target proteins. SUMOylation of some (but not all) targets require help of the one of several E3 ligases that bring target protein and Ubc9-SUMO into proximity facilitating the transfer of SUMO. (B) Change in genes and transposons expression upon SUMO KD in the ovary. (Left) Gene and transposon expression

*Figure 3 continued on next page*

**Figure 3 continued**

in RNA-seq data from smt3 KD and control ovaries. Germline-specific knockdown of smt3 was induced by small hairpin driven by maternal-tubulin-Gal4 driver; shRNA against the white gene was used as a control. Genes that change significantly (qval <0.05, LRT test, sleuth [**Pimentel et al., 2017**]) 2-fold and above are highlighted. R1, R2 transposons and *smt3* gene are labelled. (Right) Boxplot shows the distribution of fold changes in *smt3* KD versus control ovaries of genes (RefSeq) and transposons (RepBase). Genes with infinite fold change values (zero counts in control ovaries) are not shown. One is added to log2-transformed TPM values to enable the visual display of genes that are not expressed in a given condition. (C-E) SUMO depletion leads to upregulation of R1 and R2 transposons (C) RNA-seq signal coverage along the R1 and R2 consensus sequences (RepBase) from control (shW) and SUMO-depleted (shSmt3) ovaries (r1 and r2 indicate two biological replicates). Data is normalized to total reads mapping to the genome. (D) RT-qPCR analysis of R1 and R2 expression normalized to rp49 mRNA from SUMO-depleted (shSmt3) and control (shW) ovaries. PCR amplicons are within the R1 and R2 sequence (R1 and R2) or spans the junction between the 28S rRNA and the transposon (rRNA-R1 and rRNA-R2). Error bars indicate standard deviation of three biological replicates. Statistical significance is estimated by two-tailed Student's t-test; *p<0.05, **p<0.01, ***p<0.001. (E) RT-qPCR analysis of R1 and R2 expression normalized to rp49 mRNA in S2 cells upon knockdown of SUMO (dsSmt3), the E2 SUMO ligase Ubc9 (dsUbc9) or control (dsGFP) by double strand RNA. Error bars indicate standard deviation of three biological replicates. Statistical significance is estimated by two-tailed Student's t-test; **p<0.01, ***p<0.001.

The online version of this article includes the following figure supplement(s) for figure 3:

**Figure supplement 1.** R1 and R2 are strongly up-regulated upon SUMO but not Su(var)2–10 depletion.

---

rDNA was detected in almost all nuclei upon Smt3 KD (**Figure 4C**). Second, depletion of SUMO increased the signal intensity per nucleus: after Smt3 KD the signal increased 35 ~ 37 fold for interrupted transgenes and ~2.5 fold for the intact transgene (**Figure 4D**). Consistent with the RT-qPCR results, intact and interrupted transgenes showed similarly high expression level upon SUMO KD. Thus, while interrupted units are preferentially silenced in wild-type cells, loss of SUMO eliminates differential repression.

To test the role of SUMO in expression of endogenous rDNA units we measured pre-rRNA levels by RT-qPCR using primers that amplify the external transcribed spacer (EST) portion. Depletion of Smt3 led to 2.5-fold increase in pre-rRNA levels both in ovarian germ cells and in S2 cells (**Figure 4E**), demonstrating that SUMO controls expression of endogenous rDNA arrays.

## The role of heterochromatin in SUMO-dependent rDNA repression

SUMO-dependent repression of rRNA might be caused by co-transcriptional degradation of pre-rRNA or by suppression of Pol I transcription. To discriminate between these possibilities, we employed ChIP to measure the presence of RNA pol I on intact and interrupted transgenes. We generated transgenic flies that express GFP-tagged Rpl135, an essential subunit of Pol I. As expected, GFP-tagged Rpl135 localized to the nucleolus (**Figure 5—figure supplement 1A**). Rpl135 ChIP-qPCR using a GFP antibody showed enrichment of Pol I on the ETS portion and on 18S of native rDNA units (**Figure 5—figure supplement 1B**). Enrichment of Pol I was also detected on promoters of the rDNA transgenes. Importantly, depletion of SUMO caused 2.5 to 4-fold increase in the levels of Pol I on intact and interrupted transgenes (**Figure 5A**), indicating that the SUMO-dependent pathway regulates rDNA transcription. rDNA arrays are embedded in heterochromatin on the X and Y chromosomes, a genomic compartment characterized by high level of the H3K9me3 mark, which is linked to repression of protein-coding genes transcribed by RNA pol II. H3K9me3 was proposed to play a role in rDNA silencing (**Santoro et al., 2002**). To study whether the SUMO pathway regulates rDNA expression through deposition of H3K9me3 mark, we performed H3K9me3 ChIP-seq in the fly ovary upon Smt3 knockdown. We found that endogenous rDNA units and R1 and R2 sequences are indeed enriched in H3K9me3 mark in wild-type flies, however, no change in H3K9me3 occupancy was observed upon SUMO depletion (**Figure 5B**). We used independent ChIP-qPCR to compare the levels of heterochromatin mark on R1 and R2 transposons to other hetero- and euchromatin regions. These experiments confirmed that R1 and R2 sequences are heterochromatic. The level of H3K9me3 was not affected by Smt3 KD on R1 and decreased by 36.7% on R2 (**Figure 5C**).

Next, we analyze the presence of H3K9me3 mark on rDNA transgenes. The level of the mark was ~4 fold lower on transgenic compared to endogenous rDNA copies and only slightly higher compared to control euchromatin region (**Figure 5D**). Heterochromatin mark levels were similar on intact and interrupted rDNA transgenes, and they did not decrease upon SUMO knockdown. Taken together, these results indicate that the levels of H3K9me3 mark on endogenous and transgenic rDNA do not show clear correlation with silencing.

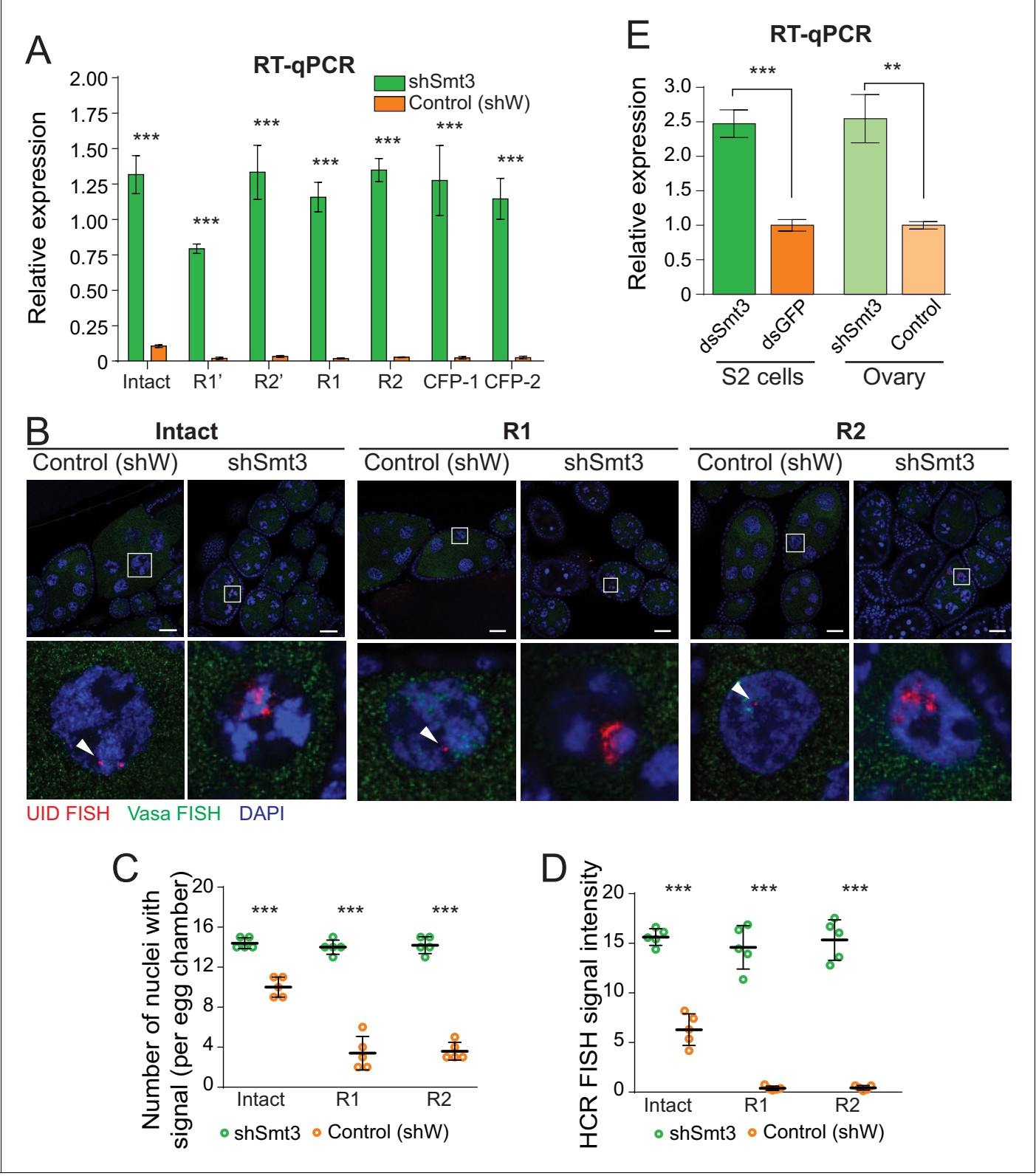

**Figure 4.** SUMO depletion leads to derepression of both interrupted and intact rDNA transgenes. (A) SUMO knockdown depresses rDNA transgenes. Expression of rDNA transgenes in ovaries as measured by RT-qPCR upon germline knockdown of SUMO (shSmt3) and in control (shW). Data shows three biological replicates normalized to rp49 mRNA. Error bars indicate standard deviation of xxxx biological replicates. Statistical significance is estimated by two-tailed Student's t-test; ***p<0.001. (B) Transgenic pre-rRNA expression is repressed by SUMO as detected by HCR-FISH Expression

*Figure 4 continued on next page*

*Figure 4 continued*

of rDNA transgenes in nurse cells is detected upon germline-specific knockdown of SUMO (shSmt3) or control (shWhite) gene using a probe against the UID sequence (Red). Vasa mRNA (green) is detected in parallel. Scale bar: 10 μm. Bottom row shows magnification of individual boxed nuclei. (C) rDNA transgenes are expressed in more nuclei upon SUMO KD as detected by HCR-FISH Shown is the number of nurse cell nuclei with positive HCR-FISH signal per egg chamber (out of 15 nuclei per stage 7/8 chamber). Each circle represents data from one egg chamber. Both intact and interrupted transgenes are expressed in all nuclei upon SUMO knockdown. Error bars indicate the standard deviation of 5 egg chambers. Statistical significance is estimated by two-tailed Student's t-test; ***p<0.001. (D) Transgenic rDNA HCR-FISH signal is increased upon SUMO KD The total intensity of HCR-FISH signal was measured in individual nuclei that have positive signal. Error bars indicate the standard deviation of 5 nuclei. Statistical significance is estimated by two-tailed Student's t-test; ***p<0.001. (E) SUMO knockdown increases pre-rRNA expression. Expression of pre-rRNAs were measured by RT-qPCR using primers that target ETS region normalized to *rp49* mRNA. Germline-specific knockdown of SUMO (shSmt3) or control (shW) gene was induced by small hairpin driven by maternal-tubulin-Gal4 driver. In S2 cells, knockdown of SUMO (dsSmt3) or control (dsGFP) was induced by double stranded RNA. Error bars indicate the standard deviation of three biological replicates. Statistical significance is estimated by two-tailed Student's t-test; **p<0.01, ***p<0.001.

To further explore the role of heterochromatin marks in repression of rDNA we used RNAi to knock-down three histone methyltransferases, *SetDB1*, *Su(var)3–9* and *G9a*, that are together responsible for all mono-, di- and trimethylation of H3K9 in *Drosophila* (*Supplementary file 4*). Depletion of *SetDB1* and *G9a* had no effect on R1 and R2 expression, while depletion of *Su(var)3–9* caused ~2.5 fold change (*Figure 5E*). Simultaneous knockdown of all three histone methyltransferases had the similar effect as depletion of *Su(var)3–9* alone indicating no redundancy. Thus, knockdown of H3K9 methyltransferases induces mild derepression of R1 and R2 compared to strong upregulation upon *smt3* and *Ubc9* RNAi.

## SUMOylation of nucleolar proteins involved in rDNA expression

Proteomic studies in *Drosophila* identified several hundred SUMOylated proteins (*Handu et al., 2015*; *Nie et al., 2009*). However, the list is likely to be even longer considering frequency of the SUMOylation consensus (ΨKxE/D) in the fly proteome and technical difficulties of detecting SUMOylation that often affects only a small fraction of target protein molecules (*Hay, 2005*). Many chromatin proteins, including histones, are substrates of SUMOylation (*Shiio and Eisenman, 2003*; *Nathan et al., 2003*) suggesting a possibility that rDNA repression might be caused by SUMOylation of chromatin proteins on rDNA sequences. To explore this possibility, we analyzed previously published SUMO ChIP-seq data (*Gonzalez et al., 2014*). This analysis revealed enrichment of SUMO at rDNA unit and R2 sequences as well as at the 5' regions of the R1 transposon (*Figure 6A*) indicating that SUMOylated proteins are indeed enriched on rDNA chromatin.

To find if specific proteins involved in rDNA transcription and nucleolar function are SUMOylated we employed sensitive SUMOylation assay. GFP-tagged candidate proteins were co-expressed together with FLAG-SUMO in S2 cells followed by their immunoprecipitation and detection of SUMOylation. The assay successfully detected SUMOylation of CTCF, conserved Zn-finger protein involved in high order chromatin organization that is known target of SUMOylation (*MacPherson et al., 2009*; *Guerrero and Maggert, 2011*). Out of 7 other candidates we detected SUMOylation of three proteins including Udd and CG3756, two proteins involved in RNA pol I dependent transcription of rDNA (*Figure 6B*). As expected from previous observations (*Hay, 2005*), only small fraction of target proteins was modified. Thus, the limited screen of selected nucleolar proteins suggests that potentially many more proteins involved in rDNA transcription might be SUMOylated.

To further explore this possibility, we retrieved all *D. melanogaster* proteins associated with Gene Ontology term 'nucleolus' (GO:0005730 and child terms, n = 243 Flybase-annotated genes) and searched for the SUMOylation consensus (ψKxE/D) in their sequences. This analysis showed that 77% of nucleolar genes have products harboring a SUMOylation consensus. In addition, 68% (19 out of 28) genes associated with the biological process 'transcription by RNA polymerase I' (GO:0006360) have a SUMOylation consensus. Proteomic approach was previously used to comprehensively identify SUMOylated proteins in *Drosophila* (*Handu et al., 2015*). Ontology analysis revealed that 27 out of 823 SUMOylated proteins are associated with the GO term 'nucleolus' demonstrating significant enrichment of nucleolar proteins among SUMOylated proteins (BH = adjusted p-value<0.01) (*Supplementary file 3*). Taken together, the results of experimental and

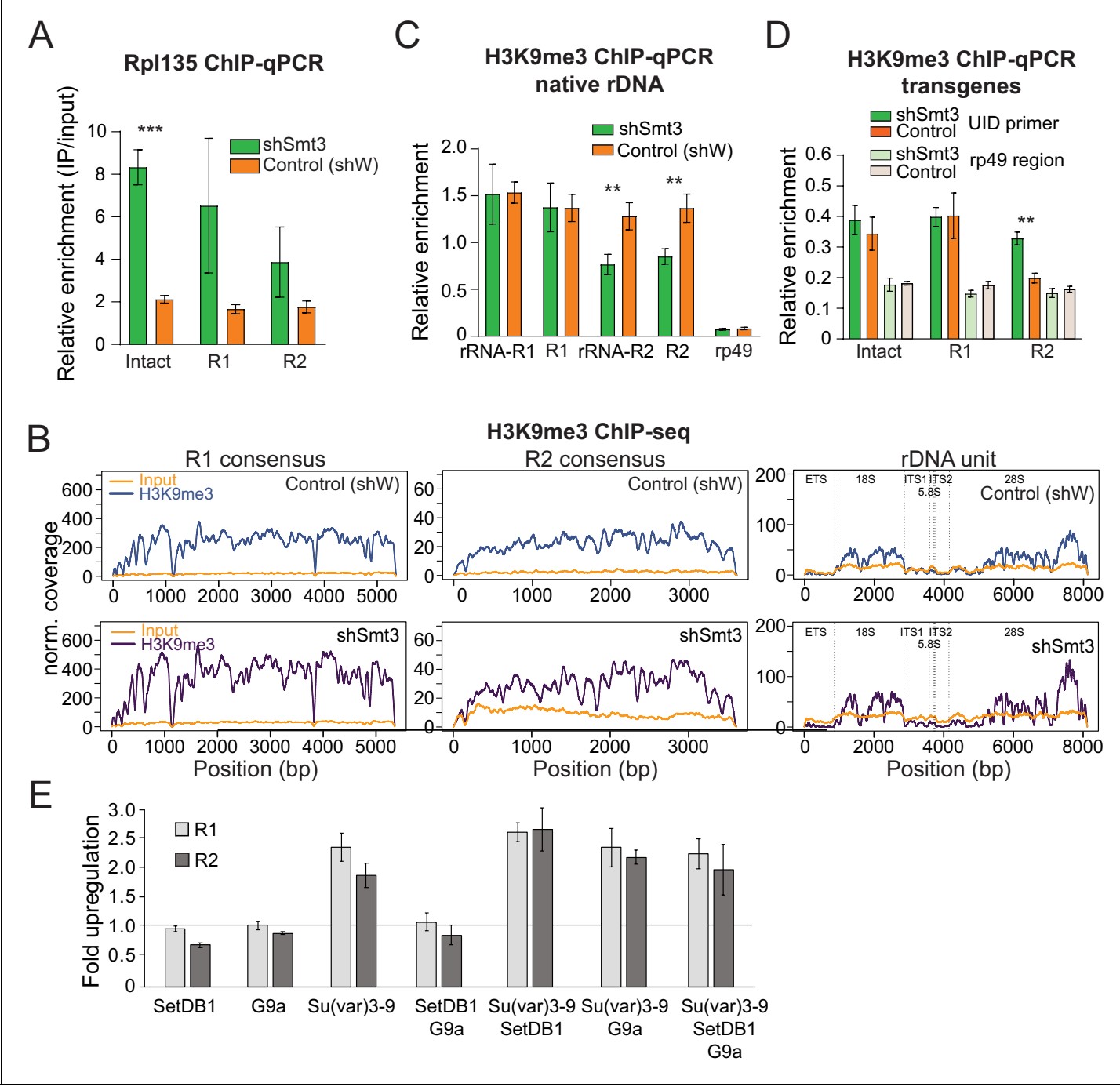

**Figure 5.** SUMO KD does not affect H3K9me3 enrichment over rDNA and R1/R2. (**A**) Pol I occupancy increases over rDNA transgenes upon SUMO KD Germline-specific knockdown of SUMO (shSmt3) or control (shW) gene was induced by small hairpin driven by maternal-tubulin-Gal4 driver. ChIP of Rpl135-GFP using a GFP antibody for pull-down was followed by qPCR analysis using UID-specific primers. Data were normalized using a sequence mapping to a gene-poor region. Error bars indicate standard deviation of three biological replicates. Statistical significance is estimated by two-tailed Student's t-test; ***p<0.001. (**B-C**) H3K9me3 enrichment over native rDNA and the R1 and R2 transposons sequences is unaffected by SUMO KD. Germline knockdown was induced by small hairpin driven by maternal-tubulin-Gal4 driver (**B**) H3K9me3 ChIP-seq signal and corresponding input coverage across the R1 and R2 consensus sequences (RepBase), and the rDNA unit (***Stage and Eickbush, 2007***) from control (shW) and SUMO-depleted (shSmt3) ovaries. Data is normalized to total reads mapping to the genome. (**C**) H3K9me3 ChIP-qPCR using primers to native 28S rDNA interrupted with R1 and R2 insertions as in ***Figure 3D***. ChIP to input enrichment is normalized to the region that has high level of H3K9me3 mark (chr2R: 4,141,405–4,141,502). Error bars indicate the standard deviation of three biological replicates. Statistical significance is estimated by two-tailed Student's t-test; **p<0.01. (**D**) H3K9me3 mark measured on rDNA transgenes by ChIP-qPCR. H3K9me3 ChIP-qPCR analysis of rDNA transgenes in

*Figure 5 continued on next page*

*Figure 5 continued*

control (shW) and SUMO-depleted (shSmt3) ovaries using primers to the UID sequence and RP49 (control region), normalized to H3K9me3-enriched region (chr2R: 4,141,405 ~ 4,141,502). Error bars indicate the standard deviation of three biological replicates. Statistical significance is estimated by two-tailed Student's t-test; **p<0.01. (E) Impact of knock-down of H3K9 methyltransferases on R1 and R2 expression. Three H3K9 methyltransferases were depleted in S2 cells using RNAi individually and in combination. Expression of R1 and R2 transposons was measured by RT-qPCR and normalized to *rp49* mRNA. Shown is fold upregulation of R1 and R2 expression upon knock-down compared to control (double-stranded RNA against eGFP gene). Expression levels were measured in three biological replicates. Statistical significance is estimated by two-tailed Student's t-test; *p<0.05, **p<0.01, ***p<0.001. Information about efficiency of RNAi KD is shown in **Supplementary file 4**.

The online version of this article includes the following figure supplement(s) for figure 5:

**Figure supplement 1.** Validation of Rpl135-GFP expression, localization in the nucleolus and binding to rDNA.

computational analyses indicate that many proteins involved in nucleolar function and rDNA transcription are SUMOylated.

To find proteins involved in SUMO-dependent rDNA repression we used RNAi to knock-down candidate genes and monitor expression of R1 and R2 transposons in S2 cells. We composed a list of proteins that have nucleolar function or involved in rDNA transcription and are SUMOylated according to published (*Handu et al., 2015*) and our own (Ninova, unpublished) mass-spec data (*Supplementary file 4*). We also selected proteins involved in SUMO pathway such as E3 SUMO ligases and SUMO isopeptidases. Out of 25 tested genes, knockdown of five (*Ulp1*, *CG13773 CG3756*, *Fib*, *mbm*) caused increase in expression of R1 and R2, including SUMO isopeptidase Ulp1 (*Figure 6C*). None of tested E3 SUMO ligases scored positive. This result might be explained by potential redundancy between tested E3 ligases. Alternatively, SUMO-dependent rDNA repression requires yet unknown E3 ligase or involves direct transfer of SUMO by E2 ligase Ubc9. The RNAi screen also revealed that repression requires two proteins that contribute to RNA polymerase I activity: CG13773 and CG3756. However, the magnitude of R1/R2 upregulation was rather modest (3 to 30-fold) compared to dramatic (>1,000 fold) derepression observed upon SUMO knockdown. Overall, the results of RNAi screen further support the role of SUMO pathway in rDNA regulation and suggest that multiple proteins including several SUMOylated components of RNA pol I complex are involved in this process.

To directly test if local SUMOylation of chromatin proteins in vicinity of rDNA promoter lead to repression we recruited E2 SUMO ligase Ubc9 to rDNA transgene. Ubc9 enzyme catalyzes the transfer of SUMO that is covalently linked to it to many proteins that harbor simple SUMOylation motif (*Johnson and Blobel, 1997*; *Figure 3A*). We generated transgenic flies with intact rDNA harboring 16-nt hairpin sequence that can be irreversibly bound by inactive version of bacterial Csy4 nuclease (*Lee et al., 2013*; *Figure 6D*). Expression of Ubc9 fused to Csy4 lead to 4.7-fold decrease in transgene expression compared to control flies suggesting that increase in local SUMOylation lead to further rDNA repression (*Figure 6D*).

## Discussion

To satisfy the high demand for rRNAs – essential components of ribosomes - genomes of most organisms contain multiple identical rDNA genes. However, studies in many eukaryotic species paradoxically demonstrated that only a fraction of available rDNA genes are expressed, while other rDNA units with apparently identical sequence are inactive (*Conconi et al., 1989*; *Morgan et al., 1983*; *Sogo et al., 1984*). rDNA repression was linked to rDNA stability, prevention of recombination and preserving nucleolar structure (*Espada et al., 2007*; *Sinclair et al., 1997*; *Oakes et al., 2006*). Differential expression of ribosomal RNA genes represents an ultimate case of epigenetic regulation: identical DNA sequences have drastically different expression levels within a single cell and these expression states are propagated through multiple cellular divisions.

Studies of rDNA repression are hampered by the fact that hundreds of rDNA units are present in the genome with almost identical sequence, which cannot be reliably discriminated (*Ganley and Kobayashi, 2007*). To circumvent this problem and understand regulation of rDNA expression, we used molecularly tagged rDNA transgenes inserted in a heterologous locus in the *D. melanogaster* genome. Insertion of a short unique sequence into the 5'-external transcribed spacer (ETS) of rDNA

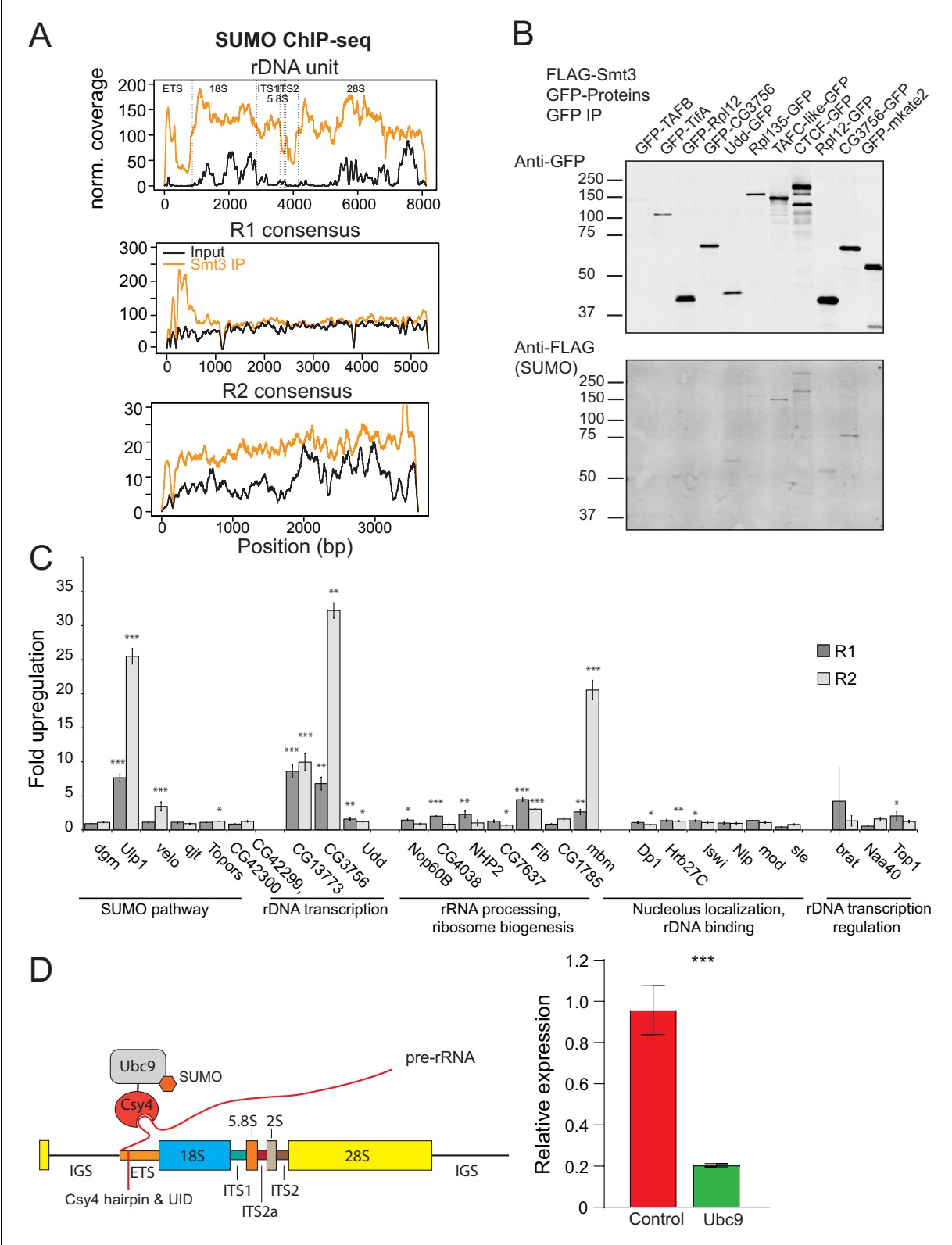

**Figure 6.** SUMOylated proteins. (**A**) SUMO ChIP-seq profile on rDNA and R1 and R2 consensus sequences. SUMO ChIP-seq signal over the R1 and R2 consensuses and the rDNA unit as in (4A). ChIP-seq data is from *Gonzalez et al., 2014*. (**B**) SUMOylation of proteins involved in rDNA transcription and nucleolar function. GFP-tagged proteins were co-expressed with FLAG-SUMO in S2 cells. SUMOylation was detected after immunoprecipitation with anti-GFP antibodies. Several proteins were tagged at either N- or C-terminus. In case of CG3756, only C-terminally tagged protein appears to be

*Figure 6 continued on next page*

*Figure 6 continued*

SUMOylated. (C) RNAi screen for genes involved in R1 and R2 repression in S2 cells. After knock-down of gene expression using RNAi expression of R1 and R2 transposons was measured by RT-qPCR and normalized to *rp49* mRNA. Shown is fold upregulation of R1 and R2 expression upon knock-down compared to control (double-stranded RNA against eGFP gene). Expression levels were measured in three biological replicates. Statistical significance is estimated by two-tailed Student's t-test; *p<0.05, **p<0.01, ***p<0.001. Information about genes and efficiency of RNAi KD is shown in *Supplementary file 4*. (D) Tethering of E2 SUMO ligase Ubc9 suppresses rDNA transgene. (Left) The scheme of tethering experiment. The rDNA transgene was modified by insertion of binding site for Csy4 RNA-binding protein into ETS. E2 SUMO ligase Ubc9 was fused with Csy4 protein and co-expressed with rDNA transgene. (Right) The results of RT-qPCR analysis of rDNA transgene expression upon Ubc9 tethering.

did not interfere with its expression and allowed us to monitor expression and chromatin structure of the rDNA transgenes and to discriminate it from the native rDNA units.

In many organisms, including *Drosophila* native rDNA clusters are located within constitutive heterochromatin, repeat-rich and gene-poor regions that are transcriptionally silent (*Németh and Längst, 2011*; *Fujiwara et al., 1998*). In agreement with previous studies (*Karpen et al., 1988*), our results show that individual ribosomal RNA units inserted in a heterologous genomic position outside of the native rDNA cluster can be transcribed by Pol I and processed (*Figures 1C* and *2C*, *Figure 5—figure supplement 1*). Insertion of an individual rDNA unit in a euchromatic region in *Drosophila* was shown to be transcribed and to recruit nucleolar proteins to form nucleoli that are morphologically and functionally similar to endogenous nucleoli (*Karpen et al., 1988*). Furthermore, individual rDNA units in heterologous genomic locations are able to partially rescue the bobbed phenotype caused by decreased number of endogenous rDNA units, indicating that transgenic rRNA is functional and incorporated into ribosomes. Similarly, individual rDNA insertions into different genomic sites in *S. cerevisiae* were shown to be competent in assembly of the RNA Polymerase I complex and production of ribosomal RNA (*Oakes et al., 2006* ). Thus, our results corroborate previous studies that suggest that the repeated structure and genomic location of rDNA loci are not required for their transcription.

Previous studies suggested that rDNA units that are interrupted by insertion of R1 and R2 retrotransposons, which integrate into specific sites in 28S rDNA in *Drosophila* are selectively repressed (*Eickbush and Eickbush, 2003*; *Jolly and Thomas, 1980*; *Kidd and Glover, 1981*; *Long et al., 1981*; *Long et al., 1981*). Variable levels of repression were also observed upon integration of a non-transposon sequence in 28S rDNA in a native rDNA cluster (*Eickbush and Eickbush, 2003*). The molecular mechanism by which interrupted rDNA units are repressed and its link to silencing of intact rDNA remained poorly understood. Our results indicate that integration of a sequence other than transposons can induce rDNA repression. Interestingly, extremely truncated R2 copies were shown to be actively expressed, indicating that rDNA with very short insertions can escape silencing (*Eickbush and Eickbush, 2003*). However, although the length of CFP (720 bp) is shorter than the full-length R1 and R2 transposons (5.4 and 3.6 kb, respectively), it is sufficient to induce repression. Integrating a heterologous sequence into rDNA also induced silencing in mammals, which lack transposons that specifically integrate into rDNA. Integration of the human growth hormone gene into rat ribosomal locus caused deletion of the ribosomal sequence and silencing of rDNA (*Henderson and Robins, 1982*). Thus, repression of rDNA copies interrupted by insertions of heterologous sequences seems to be an evolutionary conserved process that prevents production of aberrant rRNA, which might interfere with proper ribosome assembly.

Detection of transcripts from intact and interrupted rDNA transgenes in individual cells revealed large cell-to-cell variability in their expression. Although intact transgenes are expressed in a larger fraction of cells than interrupted transgenes, they are still not expressed in all cells, indicating that intact rDNA units also undergo silencing, albeit less frequently than interrupted units. It should be noted that expression of transgenes was studied in nurse cells that endoreplicate their DNA and cell-to-cell variability might be caused by differences in endoreplication of transgenic locus in individual cells. As we cannot assess expression of individual rDNA genes in endogenous clusters, it is not clear if they also show cell-to-cell variability.

In search of the molecular mechanism of rDNA repression we found that SUMO depletion lead to dramatic increase in the expression of R1 and R2 transposons, which integrate in native rDNA clusters (*Figure 3B–E*). Potent derepression of R1 and R2 was observed only upon SUMO depletion, but not in piRNA pathway mutants (*Ninova et al., 2020a*) or upon depletion of E3 SUMO ligase Su(var)

2–10 (*Ninova et al., 2020a*) indicating that SUMO plays a special role in repression of rDNA-targeting transposons that is independent of the piRNA pathway. Furthermore, depletion of SUMO also activates rDNA transgenes indicating that they are repressed by the same mechanism that silences native rDNA units (*Figure 4A*). SUMO KD increased expression of intact transgenes and released their repression in individual cells indicating that the same SUMO-dependent pathway is responsible for repression of both interrupted and intact rDNA. At present, we do not know if expression of individual rDNA genes in endogenous clusters is variable between cells. Unlike transgenes located in euchromatin, endogenous rDNA genes are arranged in large arrays of tandem repeats in heterochromatin. Thus, unnatural genomic environment might be responsible for transgenes expression variability. However, the finding that disruption of SUMO pathway activates endogenous rDNA and at the same time eliminates variability of transgene expression suggests that this variability might be linked to natural process of SUMO-dependent rDNA repression.

## Mechanism of SUMO-dependent rDNA repression

SUMO-dependent repression correlates with the levels of nascent pre-rRNA (*Figure 4B–D*) and decreased Pol I occupancy at rDNA promoters (*Figure 2C*) suggesting that repression acts at the level of transcription. How the presence of an insertion within the body of the rDNA unit lead to decreased transcription at the promoter remains to be understood. In mammals repression of rDNA units was shown to correlate with the presence of CpG DNA methylation and repressive histone marks near the rDNA promoter (*Bird et al., 1981*; *Coffman et al., 2005*; *Conconi et al., 1989*; *Dammann et al., 1993*; *Flavell et al., 1988*; *Earley et al., 2006*; *Li et al., 2006*; *Santoro and Grummt, 2001*; *Santoro et al., 2002*; *Zhou et al., 2002*). Furthermore, the chromatin remodeling complexes NoRC and NuRD were shown to be involved in rDNA repression probably by altering accessibility of the rDNA promoter to chromatin repressors such as DNA methyltransferases, histone methyltransferases and deacetylases (*Santoro et al., 2002*; *Strohner et al., 2001*; *Xie et al., 2012*; *Zhou et al., 2002*). Unlike mammals *Drosophila* lack DNA methylation (*Urieli-Shoval et al., 1982*), so this mechanism plays no role in rDNA repression in flies. Our results indicate that native (but not transgenic) rDNA units are enriched in H3K9me3, a histone mark associated with repression of genes transcribed by Pol II. However, the level of H3K9me3 remained high on rDNA units upon SUMO depletion (*Figure 5*). Furthermore rDNA transgenes inserted in heterologous genomic locus are repressed by SUMO-dependent mechanism though they have low level of H3K9me3 mark compared to endogenous rDNA. On the other hand, we observed mild increase in expression of R1 and R2 retrotransposons upon knock-down of *Su(var)3–9* H3K9 methyltransferase indicating that heterochromatin marks play a role in regulation of rDNA expression (*Figure 5E*). Taken together, our results indicate that H3K9 methylation might contribute to rDNA silencing along with other pathways, however, it cannot be the only component of SUMO-dependent repression mechanism.

Large number of nuclear proteins are SUMOylated and SUMO is required for many nuclear processes (*Nacerddine et al., 2005*; *Zhao and Blobel, 2005*) including different stages of ribosome maturation (*Heun, 2007*; *Takahashi et al., 2008*; *Finkbeiner et al., 2011*) and transcriptional repression of protein-coding genes (*Gill, 2005*; *Verger et al., 2003*). Therefore, SUMO might play two different – though not necessarily mutually exclusive – functions in rDNA silencing. First, repression might depend on SUMOylation of proteins directly involved in rDNA silencing. Alternatively, depletion of SUMO might influence rDNA expression indirectly by changing expression of genes involved in repression. In agreement with known role of SUMO in Pol II transcription, its depletion in female germline caused changes in gene expression. We observed statistically significant >2 fold up-regulation of ~4% (323) and down-regulation of ~1.5% (138) of the analyzed genes (qval <0.05, likelihood ratio test). However, the vast majority of affected genes changed within 2–10 fold, in stark contrast to dramatic ~1000 and~300 fold upregulation of R1 and R2 elements upon SUMO depletion (*Figure 3B*). We did not find any significantly enriched GO terms associated with the up- and down-regulated gene sets upon SUMO KD related to nucleolar function and/or rRNA transcription (BH-adjusted p-value cutoff 0.05). Manual inspection showed that the only gene associated with the cellular component GO term 'nucleolus' ('GO:0005730') and offspring terms among down-regulated genes was smt3 itself. Only two out of the 323 up-regulated genes - ph-p and CG9123 - are associated with the 'nucleolus' GO term, however, neither of them has known function in rDNA expression. We did not find any matches to the biological process GO:0006360 (transcription by RNA polymerase I) and offspring. Therefore, while we cannot completely exclude a possibility of

secondary effects of SUMO depletion, there is no direct indication that expression of genes involved in rDNA transcription and nucleolar function is affected by SUMO depletion.

Several lines of evidence suggest that SUMO plays direct role in rDNA silencing through modification of one or several proteins involved in rDNA expression. First, many chromatin proteins, including histones, are substrates of SUMOylation (*Shiio and Eisenman, 2003*; *Nathan et al., 2003*) and ChIP-seq analysis revealed enrichment of SUMO at R1/R2 and rDNA sequences (*Figure 6A*). Second, our analysis of SUMOylated proteome using both targeted and unbiased approaches indicates that many nucleolar proteins, including several components of RNA pol I machinery and Udd protein previously implicated in rDNA expression (*Zhang et al., 2014*), are SUMOylated in vivo (*Figure 6B*, *Supplementary file 4*). Third, recruitment of SUMO ligase Ubc9 to rDNA transgene led to its repression revealing direct effect of local SUMOylation on rDNA expression (*Figure 6D*). Finally, local SUMOylation in proximity of gene promoters was shown to induce repression of Pol II driven transcription in Drosophila and mammals (*Ninova et al., 2020a*; *Ninova et al., 2020b*; *Stielow et al., 2008*; *Smith et al., 2011*; *Lehembre et al., 2000*; *Yang et al., 2003*; *Yang and Sharrocks, 2004*). Particularly, SUMOylation at promoters is required for repression induced by KRAB-ZFPs, the largest family of mammalian transcriptional silencers (*Li et al., 2007*). Pol I-driven rRNA promoters were found to be one of the most prominent sites of active sumoylation in human cells (*Neyret-Kahn et al., 2013*). Remarkably, inhibition of sumoylation caused upregulation of rRNA expression indicating that the role of SUMO in rDNA regulation is conserved between insects and mammals (*Neyret-Kahn et al., 2013*; *Peng et al., 2019*).

Our results suggest that SUMOylation of chromatin-associated proteins might act as a molecular mark for rDNA repression. The targets of SUMOylation involved in rDNA repression remained to be identified. Unlike SUMO depletion, knock-down of several proteins that are both SUMOylated and have known functions in rDNA expression cause relatively mild increase in R1/R2 expression (*Figure 6C*) arguing against a possibility that any single protein from this list is solely responsible for SUMO-dependent repression. However, modest R1/R2 upregulation observed upon depletion of several proteins suggest that repression might require modification of several chromatin-bound proteins rather than one specific target. Promiscuous modification of multiple proteins is in line with the findings that SUMOylation consensus is very simple and quite common in the fly proteome (*Nie et al., 2009*) and our results that show that multiple nucleolar proteins are SUMOylated. The proposal that rDNA repression depend on cumulative SUMOylation of several proteins localized on rDNA corroborated by studies in yeast that led to the ''SUMO spray'' hypothesis (*Psakhye and Jentsch, 2012*) which suggests that SUMOylation of multiple proteins localized in physical proximity is required to induce downstream response in SUMO-dependent pathway. Importantly, multiple interactions between SUMOylated proteins and proteins that bind SUMO act synergistically and thus SUMOylation of any single protein is neither necessary nor sufficient to trigger downstream processes (*Jentsch and Psakhye, 2013*; *Psakhye and Jentsch, 2012*). It will be interesting to see if SUMO promiscuously 'sprayed' on rDNA units to induce their repression.

## Materials and methods

### Fly stocks

All flies were raised at 25˚C. The stock with shRNA against *white* was obtained from Bloomington (BDSC #33623), shRNA against *Smt3* Su(var)2–10 were described in *Ninova et al., 2020a*. shRNAs were driven by the maternal alpha-tubulin67C-Gal4 driver (BDSC #7063 or #7062).

### Construction of rDNA transgenes and generation of transgenic flies

To make an rDNA unit marked with a unique sequence (UID), the full-length rDNA unit including the IGS (10.5 kb) was amplified by overlapping PCR from the DmrY22 plasmid (*Long and Dawid, 1979*) (a gift from the Dawid Lab). The UID sequence 5'-GACTCGAGTCGACATCGATGC-3' was inserted into the EST region at position +139 by overlap PCR. Overlap PCR was also used to insert R1, R2 and CFP at the R1 and R2 positions (2711 and 2648 nt) of the 28S rDNA. rDNA units were ligated into a vector that contains the *mini-white* gene and the ΦC31 attB site using Gibson Assembly (NEB). All rDNA units were inserted into genomic site 22A3 (y1 w1118; PBac{y+-attP-3B} VK00037, BDSC stock #9752) using ΦC31-mediated recombination.

To generate UASp-Rpl135-GFP transgenic flies cDNA of the Rpl135 gene was cloned into the pENTR/DTOPO (Invitrogen) entry vector and recombined into a Gateway destination vector containing the ΦC31 attB site, the *mini-white* marker, the UASp promoter and the eGFP C-terminal gateway cassette. The construct was integrated into the attP40 landing site at 25C6 (y1w67c23; P{CaryP} attP40). Transgenic flies used in this study are listed in Supplemental *Supplementary file 1*.

### RNA HCR-FISH and image analysis

The previously described HCR-FISH protocol (*Choi et al., 2018*) was used with the following modifications. Fly ovaries were dissected in cold PBS and fixed in 300 µl fixation solution (4% paraformaldehyde, 0.15% Triton X-100 in PBS) at room temperature followed by three washes with PBX (PBS, 0.3% Triton X-100) for 5 min each at room temperature. Samples were dehydrated in 500 µl 70% ethanol and permeabilized overnight at 4°C on a nutator. Samples were rehydrated in 500 µl wash buffer (2 × SSC, 10% [v/v] formamide) for 5 min at room temperature and pre-hybridized in 500 µl hybridization buffer (50% formamide, 5x SSC, 9 mM citric acid pH 6.0, 0.1% Tween 20, 50 µg/ml heparin, 1x Denhardt's solution, 10% dextran sulfate) for 30 min at 37°C. Following pre-hybridization, the hybridization solution containing 2 pmol of each probe was added and samples were incubated 12–16 hr at 37°C. Samples were washed four times with 500 µl probe wash buffer (50% formamide, 5x SSC, 9 mM citric acid pH 6.0, 0.1% Tween 20, 50 µg/ml heparin) for 15 min each at 37°C and three times with 5 x SSCT for 5 min each at room temperature. Samples were incubated in 500 µl amplification buffer (5x SSC, 0.1% Tween 20, 10% dextran sulfate) for 30 min at room temperature. 30 pmol each of hairpin H1 hairpin H2 were prepared separately by incubating at 95°C for 90 s and letting them cool to room temperature in the dark for 30 min. Samples were incubated with the hairpin solution for 12–16 hr in the dark at room temperature followed by washing with 500 µl 5x SSCT at room temperature in following order: 2 × 5 min, 2 × 30 min and 1 × 5 min. Samples were preserved on glass slides with mounting medium and imaged using a ZEISS LSM880 microscope. Probe sequences are listed in *Supplementary file 2*.

FISH signal intensity was analyzed using Fiji. For each nucleus the z-stacks of images were taken at same interval distance and the total FISH signal was calculated as a sum of the signals from each z-stack image.

### Immunofluorescence microscopy

Immunofluorescence microscopy was performed as previously described (*Hur et al., 2016*). Anti-Fibrillarin antibody (Abcam ab5821) was added to fixed ovaries at 1:500 dilution and incubated at 4°C overnight followed by incubation with 1:500 dilution of secondary Alexa fluor568 antibody. Images were acquired using the ZEISS LSM880.

### RNAi in S2 cells

The RNAi protocol was described previously (*Rogers and Rogers, 2008*). For RNAi screen in S2 cells, we used the protocol from Drosophila Genomics Resource Center (DGRC) cell-based RNAi (https://fgr.hms.harvard.edu/fly-cell-RNAi-6-well-format) scaled down to 24-well plate with subsequent changes. Primers to generate templates for dsRNA transcription are listed in *Supplementary file 2*. Double-stranded RNA were synthesized using the MEGAscript T7 Transcription Kit. Around $0.5 \times 10^6$ cells in serum-free media were plated to wells containing 5 µg of dsRNA. After bathing cells with dsRNA on room temperature for 30 min, complete media with 10% FBS was added. Cells were grown for 3 days until harvesting.

### RT-qPCR

About 20 fly ovaries were dissected, homogenized in 1 ml TRIzol (Invitrogen) and RNA was extracted and precipitated according to the manual. Reverse transcription was performed using SuperScript III (Invitrogen) with random hexamer. RT-qPCR target expression was normalized to rp49 mRNA expression. All qPCR primers are listed in *Supplementary file 2*.

### ChIP-seq and ChIP-qPCR

All ChIP experiments followed the protocol described previously (*Chen et al., 2016*) using anti-H3K9me3 antibody from Abcam (ab8898) and anti-GFP antibody from ThermoFisher (A11122) for

Rpl135-GFP. qPCR signal using primers to the respective regions was normalized to Kalahari as described previously (*Sienski et al., 2015*). All qPCR primers are listed in *Supplementary file 2*. ChIP-seq libraries were generated using the NEBNext ChIP-Seq Library Prep Master Mix Set. All libraries were sequenced on the Illumina HiSeq 2500 platform.

## RNA-seq

Total RNA was extracted from fly ovaries by using TRIzol according to the manual. Ribosomal RNA depletion was performed with the Ribo-Zero rRNA Removal Kit (Illumina). RNA-seq libraries were made using the NEBNext Ultra Directional RNA Library Prep Kit. Libraries were sequenced on the Illumina HiSeq 2500 platform.

Raw RNA-seq and ChIP-seq data for *smt3* KD in *D. melanogaster* ovaries is available on the GEO database, GSE115277. RNA-seq data from S2 cells is available on the GEO database, GSE141068.

## Bioinformatics analysis

For differential gene expression analysis, RNA-seq was pseudoaligned to the *D. melanogaster* transcriptome (RefSeq) and transposon consensuses (from RepBase, *Jurka et al., 2005*), using kallisto (*Bray et al., 2016*) with the following parameters: '–single -t 4 l 200 s 50 -b 30 –rf-stranded'. Subsequent differential expression analysis was performed with sleuth using the gene analysis option (*Pimentel et al., 2017*). Fold changes in gene expression were calculated from the average TPM between replicates in knockdown versus control ovaries. The R code (*R Development Core Team, 2018*) for this analysis is available on github at https://github.com/mninova/smt3_KD. Gene ontology (GO) analysis was performed on the genes that were significantly up- and down- regulated upon *smt3* KD (log2FC > 1 and qval <0.05 (likelihood ratio test in sleuth)), using the clusterProfiler R package (*Yu et al., 2012*) as well as the FlyMine web server, using all ovary-expressed genes that were not filtered out by sleuth, or all genes as background. In either case, no enrichment for rRNA- or nucleolar- related function was found (BH-adjusted p-value 0.05).

For genomic enrichment histograms and heatmaps in *Figure 3—figure supplement 1*, RNA-seq and ChIP-seq data were aligned to the dm3 assembly using Bowtie 0.12.17 (*Langmead et al., 2009*) with -a –best –strata -m 1 v two settings, and to TE consensus sequences from RepBase (*Jurka et al., 2005*) allowing up to three mismatches. TE consensus expression was calculated using an in-house python script as RPKM of total mapped reads to the genome. ChIP-seq data was further aligned to the rDNA unit (*Stage and Eickbush, 2007*) allowing zero mismatches. Per-base coverage of consensus sequences was calculated with bedtools (*Quinlan and Hall, 2010*) using total mapped reads as a scaling factor. Heatmaps were generated with the 'pheatmap' R package.

Gene products of *D. melanogaster* associated with the GO terms 'nucleolus' (GO:0005730) and transcription by RNA polymerase I ('GO:0006360') and respective offspring terms were retrieved using the EBI QuickGO tool, and their aminoacid sequences were retrieved from UniProt. Sequences were searched with the SUMOylation consensus pattern $\psi$Kx[E/D] where $\psi$ is a hydrophobic aminoacid and x is any aminoacid (*Rodriguez et al., 2001*; *Sampson et al., 2001*).

Experimentally identified SUMOylated proteins in S2 cells (n = 923) were retrieved from the list provided by Handu et al. GO enrichment analysis was performed using the clusterProfiler R package, encirhGO function (*Yu et al., 2012*) using all *D. melanogaster* proteins as background, and BH-adjusted p-value of 0.05.

## Acknowledgements

We thank Katalin Fejes Toth and members of the Aravin lab for discussion and comments. We appreciate the help of Maayan Schwarzkopf and Niles Pierce with HCR-FISH experiments. We thank Lynn Yi for help with bioinformatics analysis. We are grateful to Michael Buszczak and the Bloomington Stock Center for providing fly stocks, Igor Dawid for providing rDNA unit constructs. We thank Igor Antoshechkin (Caltech) for help with sequencing. MN is supported by NIH/NICHD grant (K99HD099316). This work was supported by grants from the National Institutes of Health (R01 GM097363) and by the HHMI Faculty Scholar Award to AAA.

## Additional information

### Funding

| Funder | Grant reference number | Author |
|---|---|---|
| National Institute of General Medical Sciences | GM097363 | Alexei A Aravin |
| Howard Hughes Medical Institute | Faculty Scholar Award | Alexei A Aravin |
| Ministry of Education and Science of Russian Federation | 14.W03.31.0007 | Alexei A Aravin |
| NICHD | K99HD099316 | Maria Ninova |

The funders had no role in study design, data collection and interpretation, or the decision to submit the work for publication.

### Author contributions

Yicheng Luo, Formal analysis, Investigation, Visualization, Writing - original draft; Elena Fefelova, Formal analysis, Investigation, Writing - original draft; Maria Ninova, Software, Formal analysis, Visualization; Yung-Chia Ariel Chen, Investigation; Alexei A Aravin, Conceptualization, Supervision, Funding acquisition, Validation, Methodology, Writing - original draft, Project administration, Writing - review and editing

### Author ORCIDs

Yicheng Luo https://orcid.org/0000-0003-3704-2389
Elena Fefelova http://orcid.org/0000-0002-9185-1243
Alexei A Aravin https://orcid.org/0000-0002-6956-8257

### Decision letter and Author response

Decision letter https://doi.org/10.7554/eLife.52416.sa1
Author response https://doi.org/10.7554/eLife.52416.sa2

## Additional files

### Supplementary files

• Supplementary file 1. *Drosophila melanogaster* stocks.

• Supplementary file 2. Primers.

• Supplementary file 3. Cellular component (CC) GO analysis of SUMOylated proteins identified in S2 cells. The table shows results from GO enrichment analysis for the cellular component on SUMOylated proteins identified by *Handu et al., 2015*, performed using the enrichGO function of the clusterProfiler package (*Yu et al., 2012*).

• Supplementary file 4. RNAi screen for genes involved in R1 and R2 repression in S2 cells For each gene the information column describes it known or predicted function and nucleolus localization. Association with GO term 'nucleolus' (GO:0005730) and its child terms is shown according to Gene Ontology analysis. The presence of SUMOylation motif(s) was indicates the presence of consensus SUMOylation sequence (ψKxE/D) in protein sequence. Expression of R1 and R2 transposons were measured by RT-qPCR in S2 cells after knock-down using double-stranded RNA using primers listed in *Supplementary file 2*. Shown is fold change in R1 and R2 levels upon knock-down of corresponding gene compared to control KD (double-stranded RNA against eGFP gene). Knockdown efficiency shows the depletion of target gene as measured by RT-qPCR. Expression levels were measured in three biological replicates and normalized to rp49 mRNA. Statistical significance is estimated by two-tailed Student's t-test; *p<0.05, **p<0.01, ***p<0.001.

• Supplementary file 5. Numerical data from RNA-seq analysis by sleuth.

• Transparent reporting form

## Data availability

Sequencing data have been deposited in GEO under accession codes GSE141068 and GSE115277. Other data generated or analysed during this study are included in the manuscript and supporting files.

The following datasets were generated:

| Author(s) | Year | Dataset title | Dataset URL | Database and Identifier |
|---|---|---|---|---|
| Luo Y, Fefelova E, Ninova M, Chen YA, Aravin AA | 2020 | Repression of damaged and intact rDNA by the SUMO pathway | https://www.ncbi.nlm.nih.gov/geo/query/acc.cgi?acc=GSE141068 | NCBI Gene Expression Omnibus, GSE141068 |
| Ninova M, Chen YA, Godneeva B, Rogers A, Luo Y, Tóth KF, Aravin AA | 2019 | The SUMO ligase Su(var)2-10 links piRNA-guided target recognition to chromatin silencing | https://www.ncbi.nlm.nih.gov/geo/query/acc.cgi?acc=GSE115277 | NCBI Gene Expression Omnibus, GSE115277 |

The following previously published dataset was used:

| Author(s) | Year | Dataset title | Dataset URL | Database and Identifier |
|---|---|---|---|---|
| Gonzalez I, Mateos-Langerak J, Thomas A, Cheutin T, Cavalli G | 2014 | Identification of Regulators of the Three-Dimensional Polycomb Organization by a Microscopy-Based Genome-Wide RNAi Screen | https://www.ncbi.nlm.nih.gov/geo/query/acc.cgi?acc=GSE55303 | NCBI Gene Expression Omnibus, GSE55303 |

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
