## [Decision Letter]

**Acceptance summary:**

The authors investigate the observed repression of transposed (and endogenous) sequences into the rDNA locus in *Drosophila*. They show that the SUMO pathway is involved in this repression, unrelated to the piRNA pathway or histone methylation.

**Decision letter after peer review:**

Thank you for submitting your article "Repression of damaged and intact rDNA by the SUMO pathway" for consideration by *eLife*. Your article has been reviewed by three peer reviewers, one of whom is a member of our Board of Reviewing Editors, and the evaluation has been overseen by James Manley as the Senior Editor. The reviewers have opted to remain anonymous.

The reviewers have discussed the reviews with one another and the Reviewing Editor has drafted this decision to help you prepare a revised submission.

Summary:

The authors investigate the observed repression of transposed (and endogenous) sequences into the rDNA locus in *Drosophila*. They show that the SUMO pathway is involved in this repression, unrelated to the piRNA pathway or histone methylation.

While all reviewers judged this work to be solid, they also thought that there needed to be more experimental data that begins to get at the mechanism of the SUMO induced silencing.

The complete reviews are included below.

Essential revisions:

More direct analyses maximizing the use of rDNA insertions are needed to truly advance the field and to make the manuscript appropriate for publication in *eLife*, by providing more definitive support for the conclusions and elucidating which of the many potential SUMO-associated pathways and mechanisms are required.

Reviewer #1:

The authors investigate the observed repression of transposed (and endogenous) sequences into the rDNA locus in *Drosophila*. They show that the SUMO pathway is involved in this repression, unrelated to the piRNA pathway or histone methylation.

The data and conclusions of this work are solid and convincing. It is interesting and intriguing that Sumo is involved in the repression of the transposed elements, but how does this work? One might expect some analysis of the pol I dependent process, presumably at the promoter, perhaps the modification of a factor involved in repression. They report an increase in polymerases upon Sumo KD, so this likely indicates that initiation was affected. The almost three orders of magnitude increase in expression upon Sumo KD is an observation begging for more experiments since it's so dramatic.

The authors state in that "…we achieved important progress in resolving a forty-year old mystery in epigenetic regulation of rDNA expression." This appears to be a bit of an overstatement: we know SUMO is involved, somehow, but it's far from resolving a mystery. They feel that getting these results out in press would engage more research into this repressive mechanism. Fair enough but I think the burden is on them to lead the way, having made this observation. They feel that it will takes years of research to do this, but it seems that a bit more substance could be included with some additional work (what are the potential targets? Can RNAi determine if they are part of the pathway?).

That being said, I do not oppose its publication, since it does identify a factor in the repression of ribosomal RNA genes, so progress has been made. But I'm not that enthusiastic either since it's an observation begging for some more molecular context.

Reviewer #2:

In the manuscript "Repression of damaged and intact rDNA by the SUMO pathway", Luo and colleagues show that the SUMO pathway plays a role in regulating the expression of rDNA, in particular damaged rDNA in *Drosophila*. They find that the transcriptional repression of rDNA with insertions of retrotransposons R1 and R2, which preferentially insert into rDNA is independent of the piRNA pathway and requires the SUMO pathway. The experiments manuscript is very well-written and a pleasure to read and the experiments presented are largely unimpeachable and support the message.

Nevertheless, enthusiasm for the manuscript is slightly dampened considering that the authors do not attempt to gain a deeper understanding of the phenomenon they found.

At least some of the following points may have been expected to find their way into the initial report (the first two points are technical and the rest goes beyond what the manuscript shows):

1) Knockdown of SUMO would be expected to have broad impact on gene regulation with – what is the evidence that in the chosen experimental conditions only R1 and R2 transposons and rDNA transcription are directly affected and their derepression is not due to secondary effects? The RNAseqs in Figure 3 and Figure 3—figure supplement 1 are only showing transposons and are not summarizing the effect on mRNAs and other ncRNAs.

2) Figure 3B shows that Pol I occupancy on the transgenes is increased upon SUMO KD; however, the effect on the effect on intact rDNA is even stronger than on damaged rDNA, in contrast to Figure 3A, where transcriptional derepression on intact rDNA is ~5-fold lower than on damaged rDNA. Considering that the experiment in Figure 3A is crucial for the message of the manuscript claiming that SUMO-dependent pathways are sensing damaged rDNA, it would seem important to reconcile this discrepancy.

3) There are virtually no data addressing the question of how SUMO-dependent pathways sense damaged rDNA. The transgenes have their lesions only introduced at a fixed site in the 28S section of rDNA. What would happen if other sections, including the ITS and ETS were damaged? What is the minimal insert size that would be tolerated – it appears that a 29 bp insertion is already problematic.

4) There are no data included that would point to the conservation of SUMO regulation of rDNA transcription in other organisms – in particular mammals.

5) Is there any way to identify or discuss the E3-ligase responsible for the effect? Clearly it is not Su(var)2-10.

Reviewer #3:

Eukaryotic nuclei contain hundreds of copies of rDNA repeats that generate rRNA to meet the cell's metabolic needs. Despite the high levels of rRNA transcription taking place in the nucleolus, not all rDNA repeats are transcribed. A fraction of the silenced rDNA repeats in arthropods including *Drosophila* contain retrotransposons R1 and R2, which are integrated into rDNA. The mechanisms by which rDNA units, including those with R1/R2 insertions, are differentially silenced are unclear. In this manuscript, the authors create a powerful system of single rDNA transgene insertions with or without foreign sequences like R1 and R2. Using this system, they find that insertion of R1/R2 leads to rDNA silencing. Further, they conclude that SUMO induced rDNA silencing is independent of H3K9me3. This study does increase our knowledge of the components involved in rDNA silencing by identifying a role for SUMO in the process. However, there are major conclusions that are inadequately supported by the data. More direct analyses maximizing the use of rDNA insertions are needed to truly advance the field and to make the manuscript appropriate for publication in *eLife*, by providing more definitive support for the conclusions and elucidating which of the many potential SUMO-associated pathways and mechanisms are required.

Essential revisions:

1) The authors conclude that H3K9me3 is not sufficient for rDNA silencing since H3K9me3 levels are unchanged upon SUMO knockdown and associated rDNA activation. However, the evidence presented is insufficient to conclude there is no dependence on H3K9me3. First, the presented ChIP data is not convincing, e.g.H3K9me3 ChIP Seq signal is slightly higher in Figure 5A. For the ChIP-qPCR analysis of the transgene (Figure 5C), H3K9me3 levels are higher upon Smt3 KD in "Intact" (slightly) and "R2" (almost 2-fold), and p-values are not provided. Second, population measurements using ChIP-seq on polyploid fly ovaries do not capture the variation between cells or even the different rDNA units within the same cell. Thus, average K9me levels may mask individual gene/cell changes in K9me associated with silencing. Third, both me2 and me3 should be analyzed, since in fission yeast (Moazed) they are associated with different impacts. Finally, the impact of Sumo deletions, or even E2/E3 mutation, are very difficult to interpret, e.g. compared to mutating target Ks in the proteins. The observed phenotypes can be caused by disruption of any point in one or multiple pathways, including impacting complexes that increase and decrease K9me3, alter the cell cycle, replication or DNA repair, etc., potentially masking correlations between K9me and rDNA repression.

2) It is hard to conclude that repression of rDNA transgenes with insertions differs from cell-to-cell based on their analysis on the highly polyploid nurse cells in the fly ovary (Figure 2C-E), because the analysis cannot distinguish between variations arising from endoreplication vs. differential transcription (same holds for point 1, conclusions about K9me). To rule out this possibility, and to demonstrate conservation of mechanism in non-polyploid tissues, the same analysis should be performed in diploid somatic cells such as neuroblasts. This concern also holds for differences in pre-rRNA expression reported in Figure 4F upon Smt3 knockdown.

3) The authors cannot conclude that "insertion of any heterologous sequence into rDNA leads to transcriptional repression", since only one (CFP) was tested in both ovaries and carcasses. Insertions with different lengths and compositions are needed to make this conclusion. More appropriately, the results indicate that integration of a sequence other than transposons can induce rDNA repression.

4) I would like to see more information about the functionality of UID-tagged rDNA transcripts. Does the addition of the UID to the rDNA transgene alter its ETS processing kinetics, thus potentially affecting transgene expression measurements? Is the UID marked transcript inside the normal larger nucleolus or in an ectopic nucleolus?

5) There is inadequate evidence to support the claim that SUMO is required for "selective repression of damaged rDNA and intact surplus units" (final statement of the Abstract). For example, this claim is contradicted by increased expression of the single intact transgene upon SUMO knockdown (Figure 4A), indicating that SUMO does not selectively repress only "surplus" intact genes or 'damaged rDNA'.

[Editors' note: further revisions were suggested prior to acceptance, as described below.]

Thank you for resubmitting your work entitled "Repression of interrupted and intact rDNA by the SUMO pathway in *Drosophilamelanogaster*" for further consideration by *eLife*. Your revised article has been evaluated by James Manley (Senior Editor), Robert Singer (Reviewing Editor) and two reviewers.

The manuscript has been improved but there are some remaining issues that need to be addressed before acceptance, as outlined below:

Reviewer #3 thought that modifications to the text are desirable before acceptance. In particular, there are points raised about interpretation of the data, and some conclusions need to be tempered or clarified. Their detailed comments can be found below.

Reviewer #3:

The authors' revisions and new experiments improve the manuscript significantly. Even though there are many mechanistic questions left unanswered, it is important that the results show a critical role for SUMOylation in rDNA silencing. How and why SUMO causes silencing is unknown and awaits future studies. However, there are still issues with the text that need to be fixed. In most cases the authors just need to be more careful about their conclusions.

Silencing and K9me. The authors do make a clear case for no or minimal changes in overall levels of H3K9me3 upon SUMO KD. I would still be careful about making the conclusion that H3K9me3 has no role in rDNA silencing. Figure 5C reports that SUMO KD results in an 'only decreased by 36.7% on R2' in the endogenous cluster, but such a change is not trivial and could have impact, especially if the distribution across the rDNA cluster is non-random. Yes, the single rDNA transgenes are derepressed up to 1000 fold by SUMO/Ubc9 KD, and show 'only' a 2.5-fold change in rDNA transcript levels when all 3 HMTases are removed (NB: No legend for Figure 5E…and it is important to know if this figure reports expression from endogenous or transgene rDNA, not clear from the manusript text either). However, expression does increase 2.5x, and the expectation that the magnitude of the change should be similar may not be true, given the potential involvement of multiple proteins and pathways. If K9me is one but not the only component of rDNA silencing (e.g. if SUMOylation of HP1 and other proteins *not* directly involved in K9 me is required) SUMO loss would be expected to have a much larger effect. Loss of SUMO modifications on multiple proteins is also consistent with the large number of candidates identified by the screen, which could also be acting additively/synergistically: 'However, the vast majority of affected genes changed within 2-10 fold, in stark contrast to dramatic ~1000- and ~300-fold upregulation of R1 and R2 elements upon SUMO depletion (Figure 3B).'

Further, there are reasons to suspect that K9me pathways do impact repression of rDNA. E.g. most rDNA is silent and physically located in the heterochromatin outside the nucleolus. Also, the relationship between this mark and silencing is not straightforward… for coding genes embedded in heterochromatin, higher levels of K9 me are correlated positively with expression. The point here is to be careful about the interpretation and elimination of a role for K9 me pathways.

E3 SUMO ligases. Cannot conclude that 'R1/R2 repression.… is independent of Su(var)2-10 SUMO ligase.' There is precedent even in *Drosophila* for additive effects (redundancy) of SUMO E3s, so one has to look at double and triple mutants in this case as well. Thus it is premature to emphasize possible new E3s over redundancy:

"None of tested E3 SUMO ligases scored positive suggesting that SUMO dependent rDNA repression requires yet unknown E3 ligase or involves direct transfer of SUMO by E2 ligase Ubc9 that is indispensable for the repression.'

Variation in endoreplication vs. transcription. The 15 nurse cells do not endoreplicate to the same extent, and there could be even more variation in rDNA levels in these cells given that rDNA is already underreplicated relative to the euchromatin. FISH for rDNA (not RNA) would address this issue definitively. Further, and perhaps more importantly, the expression variability is interesting, but likely specific to the transgenes, since they are not tandemly repeated or in the same heterochromatic environment as the endogenous cluster. The relationship between transgene variability and whether this occurs for individual genes in the endogenous cluster should be discussed.

Finally, though not a major point, there are many remaining references to 'damaged' rDNA that should be removed, and replaced with 'interrupted' or 'inserted', to avoid reader confusion. This includes the phrase 'damaged by transposon insertions'. Damage strongly implies they are harmed…but being interrupted or silenced is only harmful if it causes cellular or organismal defects. rDNA insertions could be part of an important regulatory network that responds to metabolic changes or stress. There is evidence for such regulation, e.g. Templeton, (1980s) showed changes in the proportion of interrupted units during adaptation to changing environmental conditions, such as dessication (probably via differences in endoreplication). Needs to be looked at and validated with modern eyes and techniques, but still suggests that interrupted rDNA genes could play an important regulatory role and should not be considered 'damaged'.

(Henderson and Robins, 1982) is not in the bibliography.

---

## [Author Response]

Essential revisions:More direct analyses maximizing the use of rDNA insertions are needed to truly advance the field and to make the manuscript appropriate for publication in eLife, by providing more definitive support for the conclusions and elucidating which of the many potential SUMO-associated pathways and mechanisms are required.Reviewer #1:The authors investigate the observed repression of transposed (and endogenous) sequences into the rDNA locus in *Drosophila*. They show that the SUMO pathway is involved in this repression, unrelated to the piRNA pathway or histone methylation.The data and conclusions of this work are solid and convincing. It is interesting and intriguing that Sumo is involved in the repression of the transposed elements, but how does this work? One might expect some analysis of the pol I dependent process, presumably at the promoter, perhaps the modification of a factor involved in repression. They report an increase in polymerases upon Sumo KD, so this likely indicates that initiation was affected. The almost three orders of magnitude increase in expression upon Sumo KD is an observation begging for more experiments since it's so dramatic.The authors state that "…we achieved important progress in resolving a forty-year old mystery in epigenetic regulation of rDNA expression." This appears to be a bit of an overstatement: we know SUMO is involved, somehow, but it's far from resolving a mystery. They feel that getting these results out in press would engage more research into this repressive mechanism. Fair enough but I think the burden is on them to lead the way, having made this observation. They feel that it will takes years of research to do this, but it seems that a bit more substance could be included with some additional work (what are the potential targets? Can RNAi determine if they are part of the pathway?).That being said, I do not oppose its publication, since it does identify a factor in the repression of ribosomal RNA genes, so progress has been made. But I'm not that enthusiastic either since it's an observation begging for some more molecular context.

As suggested by the reviewer, we used several different approaches to get further insight into the molecular mechanism of SUMO-dependent rDNA repression, identify potential targets of SUMOylation and their role in rDNA expression. First, we tested if local SUMOylation of chromatin targets induces rDNA repression by tethering SUMO ligase Ubc9 to rDNA transgene. Tethering of Ubc9 lead to significant decrease in transgene expression, suggesting that local SUMOylation of chromatin targets directly triggers rDNA repression (Figure 6D).

We used two different approaches to find SUMOylated proteins involved in rDNA expression. We selected several proteins known to be involved in rRNA transcription and explore whether they are SUMOylated in S2 cells using IP/Western blot (Figure 6B). We also used the results of comprehensive mass-spec identification of SUMOylated proteins in *Drosophila* – both published (Handu et al., 2015) and our own – to find SUMOylated proteins involved in nucleolar function and rDNA transcription. Both approaches revealed that many nucleolar proteins are SUMOylated, including several components of RNA pol I complex. Importantly, Gene Ontology analysis revealed significant enrichment of nucleolar proteins among SUMOylated proteins in *Drosophila* (BH=adjusted p-value <0.01) (Supplementary file 3). Furthermore, bioinformatic analysis showed that 77% of all nucleolar proteins and 68% of all proteins involved in RNA polymerase I transcription (as defined by Gene Ontology analysis) harbor the SUMOylation consensus (ψKxE/D) in their sequences. Taken together, the results of experimental and computational analyses indicate that many proteins involved in nucleolar function and rDNA transcription are SUMOylated.

We used the list of identified proteins (that are both SUMOylated and involved in nucleolar function) and selected candidates to test their role in rDNA repression using RNAi (Figure 6C). In addition to nucleolar proteins, we explored the functions of SUMO pathway components, including several different E3 SUMO ligases. Knockdown of five genes, including several components of RNA pol I complex, caused increase in expression of R1 and R2 elements (Supplementary file 4, Figure 6C). One of newly identified factors involved in repression is SUMO isopeptidase Ulp1, further supporting the role of SUMO pathway in rDNA regulation. Interestingly, all knockdowns caused relatively mild increase in R1/R2 expression (up to 30fold) compared to drastic (~1,000-fold) upregulation upon depletion of SUMO and SUMO ligase Ubc9 (Figure 3E). Together, new results suggest that rDNA repression might depend on cumulative SUMOylation of several proteins rather than one specific target. This possibility is corroborated by studies in yeast that led to the so-called ‘SUMO spray’ model that suggests that SUMOylation of multiple proteins localized in physical proximity is necessary to trigger downstream response in several SUMO-dependent pathways (Psakhye and Jentsch, 2012). We incorporated new results and their discussion in revised manuscript.

Reviewer #2:[…]1) Knockdown of SUMO would be expected to have broad impact on gene regulation with – what is the evidence that in the chosen experimental conditions only R1 and R2 transposons and rDNA transcription are directly affected and their derepression is not due to secondary effects? The RNAseqs in Figure 3 and Figure 3—figure supplement 1 are only showing transposons and are not summarizing the effect on mRNAs and other ncRNAs.

We are thankful for this comment that prompt us to perform more detailed analysis of RNAseq data.

We analyzed global changes of gene expression upon SUMO KD and presented results on the new Figure 3B. SUMO depletion changed expression of protein-coding genes: we observed statistically significant >2-fold up-regulation of ~4% (323) and down-regulation of ~1.5% (138) genes (<0.05 BHadjusted FDR, likelihood ratio test). The vast majority of affected genes changed within 2-10 fold, in contrast to dramatic ~1000-fold upregulation of R1 and R2 elements upon SUMO depletion. Importantly, Gene Ontology analysis found no evidence that the expression of genes involved in rDNA transcription is changed: among genes down-regulated in SUMO KD the only gene associated with the cellular component GO term ‘nucleolus’ ("GO:0005730") was SUMO (smt3) itself. Out of the 323 up-regulated genes only two genes – ph-p and CG9123 – are associated with the ‘nucleolus’ GO term, however, neither of them has known function in rDNA expression. We did not find any matches to the biological process GO:0006360 (transcription by RNA polymerase I) among genes changed upon SUMO KD. Overall, the changes in protein-coding genes expression upon SUMO depletion are expected as SUMOylation is known to play important role in many cellular processes, including signaling pathways and regulation of gene expression (Stehmeier and Muller, 2009; Tripathi, et al., 2019; Yan, et al., 2010; Wang, et al., 2010). We provide the tables from differential expression analysis and gene and transposons abundances as new Supplementary file 5.

In parallel with studying effect of SUMO depletion on gene expression, we used several approaches to further explore whether SUMO is directly involved in rDNA repression. First, we recruited SUMO ligase Ubc9 to rDNA transgene and found that this led to its repression revealing direct effect of SUMOylation (Figure 6C). Second, we analyzed SUMOylated proteome using both targeted and unbiased mass-spec approaches and found that many nucleolar proteins, including several components of RNA pol I machinery, are SUMOylated (Figure 6B and Supplementary file 4). Finally, ChIP-seq analysis revealed enrichment of SUMO on rDNA sequences (Figure 5B). Combined, these results suggest that rDNA repression is mediated by SUMOylation of protein targets on rDNA chromatin, though it is difficult to completely exclude a possibility of secondary effects of SUMO depletion. We incorporated new results in revised manuscript together with the discussion of direct and secondary effects of SUMO depletion.

2) Figure 3B shows that Pol I occupancy on the transgenes is increased upon SUMO KD; however, the effect on the effect on intact rDNA is even stronger than on damaged rDNA, in contrast to Figure 3A, where transcriptional derepression on intact rDNA is ~5-fold lower than on damaged rDNA. Considering that the experiment in 3A is crucial for the message of the manuscript claiming that SUMO-dependent pathways are sensing damaged rDNA, it would seem important to reconcile this discrepancy.

Our results indicate that SUMO depletion cause derepression of both intact and damaged rDNA (Figure 4A). Similarly, Pol I occupancy is increased on both intact and damaged transgenes (Figure 5A), so there is no general discrepancy between measuring expression (RT-qPCR) and transcriptional occupancy (pol I ChIP). However, damaged rDNA is derepressed more strongly compared to intact rDNA, while ChIP-qPCR assay showed similar fold changes (3-4-fold) in Pol I occupancy for intact and damaged rDNA. We believe that the explanation for this minor discrepancy lies in different dynamic range of the two assays: the measuring transgene expression by RT-qPCR is more sensitive compared to measuring pol I occupancy by ChIP. Importantly, ChIP is known to have high level of background signal and therefore is less reliable if low amount of measured chromatin-bound protein is present. We want to note that no ChIP-grade antibodies to detect RNA pol I occupancy are available in *Drosophila* and we generated fly strain expressing tagged subunit of RNA pol I to be able to perform measurement of pol I occupancy.

Our results indicate that both damaged and intact rDNA transgenes are expressed on a similarly high level upon SUMO KD and greater derepression of damaged rDNA transgenes is due to their lower level of expression in wild-type flies. In contrast to pre-rRNA abundance, measurement of pol I occupancy by ChIP-qPCR show only minor difference between intact and damage transgenes in wild-type flies suggesting that ChIP-qPCR is not sensitive enough to detect the difference between pol I occupancy in wild-type flies when expression of both intact and damaged transgenes is low.

On another note, there are other results beyond measurement of Pol I occupancy that suggest that rDNA transgenes are regulated on the transcriptional or co-transcriptional levels. When we measured expression of rDNA transgenes using RT-qPCR (such as Figure 4A) and FISH (such as Figure 4C) we used primers and probes against ETS portion of pre-rRNA that is removed from pre-rRNA co-transcriptionally (Hughes and Ares, 1991). Therefore, changes in pre-rRNA abundance measured by these methods indicate transcriptional or co-transcriptional regulation.

3) There are virtually no data addressing the question of how SUMO-dependent pathways sense damaged rDNA. The transgenes have their lesions only introduced at a fixed site in the 28S section of rDNA. What would happen if other sections, including the ITS and ETS were damaged? What is the minimal insert size that would be tolerated – it appears that a 29 bp insertion is already problematic.

We agree with the reviewer that the question of how SUMO-dependent pathways sense damaged rDNA falls out of scope of our work and should be further studied in the future. We want to stress that our results indicate that SUMO-dependent repression affect both intact and damaged rDNA and suggest that ‘sensing’ of damaged rDNA is controlled by upstream mechanism that bias SUMO-dependent repression towards damaged copies.

We have tested insertions of four different sizes: 29 bp, 720 bp (CFP), 3607 bp (R2) and 5356 bp (R1) in two different sites (2711 nt and 2648 nt) in the 28S section of rDNA (Figure 2AB, Figure 4A). Interestingly, insertions of R1 and R2 sequences into two different sites produced different levels of repression (Figure 4 B: compare R1 and R1’ with R2 and R2’ constructs). However, CFP sequence inserted in these two locations produced very similar levels of repression (Figure 2B, CFP1 and CFP2 transgenes) indicating that (a) the nature of inserted sequence has an effect on the level of repression and (b) the same sequence inserted into two different sites produced similar effect. 29bp insertion is indeed has an effect, however, expression of rDNA with 29bp insertion is only ~2-fold lower compared to intact rDNA (while longer insertions lead to 4-7-fold repression). Therefore, decreasing the length of insert diminish repression. Considering that 29bp insertion produced relatively weak (2-fold) effect and the limits of assay precision, we believe reliable quantification the effect of even smaller insertions to find the absolute minimal insert size that is tolerated would be difficult. It is will be interesting to see if insertions in other rDNA regions such as ITS and ETS trigger repression and will plan to perform these experiments in the future.

4) There are no data included that would point to the conservation of SUMO regulation of rDNA transcription in other organisms – in particular mammals.

In our work we concentrated on understanding rDNA regulation in *Drosophila* and feel that the question of conservation of this mechanism in other organisms deserves separate studies. Unlike *Drosophila*, mammals lack transposons such as R1 and R2 that specifically insert into rDNA, making analysis of differential expression of rDNA units more difficult.

Previous studies linked SUMO pathway with rDNA stability in yeast (Liang, et al., 2017). In human cells Pol I-driven rRNA promoters were found to be one of the most prominent sites of active sumoylation (Neyret-Kahn et al., 2013). Furthermore, inhibition of sumoylation in mammalian cells caused upregulation of rRNA expression (Neyret-Kahn et al., 2013; Peng et al., 2019) indicating that the role of SUMO in rDNA regulation might be conserved between insects and mammals. We cited and discussed these findings in revised manuscript.

5) Is there any way to identify or discuss the E3-ligase responsible for the effect? Clearly it is not Su(var)2-10.

We used RNAi screen to explore the role of other components of SUMO pathway in rDNA repression, including E3 SUMO ligases. We found that in addition to SUMO (smt3) and E2 SUMO ligase (Ubc9) the repression requires SUMO isopeptidase Ulp1. However, none of the tested E3 SUMO ligases scored positive in our assay (Figure 6C). Using different approach, we showed that recruitment of E2 ligase Ubc9 to rDNA transgene induced its repression (Figure 6D). Thus, we further proved the role of SUMO pathway and particularly E2 ligase, but did not find E3 ligase involved in rDNA repression. These results suggest that SUMOylation required for rDNA repression is performed by yet-to-be identified E3 ligase or directly by Ubc9. Indeed, Ubc9 can transfer SUMO to the substrates directly, and SUMOylation of many targets does not require assistance of E3 ligases (Gareau and Lima, 2010). On the other hand, known E3 SUMO ligases are diverse in sequence and do not compose one protein family making it difficult to identify potential E3 ligases in silico. We added the new results and their discussion in revised manuscript.

Reviewer #3:[…]Essential revisions:1) The authors conclude that H3K9me3 is not sufficient for rDNA silencing since H3K9me3 levels are unchanged upon SUMO knockdown and associated rDNA activation. However, the evidence presented is insufficient to conclude there is no dependence on H3K9me3. First, the presented ChIP data is not convincing, e.g.H3K9me3 ChIP Seq signal is slightly higher in Figure 5A. For the ChIP-qPCR analysis of the transgene (Figure 5C), H3K9me3 levels are higher upon Smt3 KD in "Intact" (slightly) and "R2" (almost 2-fold), and p-values are not provided. Second, population measurements using ChIP-seq on polyploid fly ovaries do not capture the variation between cells or even the different rDNA units within the same cell. Thus, average K9me levels may mask individual gene/cell changes in K9me associated with silencing. Third, both me2 and me3 should be analyzed, since in fission yeast (Moazed) they are associated with different impacts. Finally, the impact of Sumo deletions, or even E2/E3 mutation, are very difficult to interpret, e.g. compared to mutating target Ks in the proteins. The observed phenotypes can be caused by disruption of any point in one or multiple pathways, including impacting complexes that increase and decrease K9me3, alter the cell cycle, replication or DNA repair, etc., potentially masking correlations between K9me and rDNA repression.

We thank the reviewer for detailed discussion of H3K9me3 and its role in rDNA repression. We realized that we did not describe our results properly in original manuscript leading to confusion. We changed the text in revised manuscript to better explain our results of measuring H3K9me3 (actual results remained identical to original submission). To further test the link between H3K9me2/3 and SUMO-dependent rDNA repression, we explored the roles of three histone methyltransferases, SetDB1, Su(var)3-9 and G9a, in rDNA repression. Together these enzymes are responsible for all (me2 and me3) modification of H3K9. Knockdowns of these enzymes did not lead to activation of R1 and R2 (Figure 5E) further supporting our conclusion that H3K9 methylation (both me2 and me3) is not involved in repression. We incorporated this result in revised manuscript.

We want to clarify our H3K9me ChIP results: Figure 5B (corresponding to Figure 5A in original submission) shows that H3K9me ChIP Seq signal is slightly higher upon SUMO KD compared to control flies. This effect is opposite to what is expected if H3K9me3 mark would be responsible for SUMO-dependent R1/R2 repression: in condition when R1 and R2 transposons are derepressed ~1,000-fold we see slightly higher H3K9me3 signal on their sequences. Higher levels of H3K9me3 on rDNA and R1/R2 upon SUMO KD might be potentially caused by redistribution of H3K9me3 caused by decrease of H3K9me3 on transposon-rich heterochromatin induced by failure of piRNA-guided transcriptional repression when SUMO pathway is inactivated (Ninova et al. 2020). Independent ChIP-qPCR results (Figure 5C corresponding to 5B in original submission) confirmed that native genomic R1 and R2 sequences are enriched in H3K9me3 mark, however, the level of H3K9me3 was not affected (or again slightly increased) by SUMO KD on R1 and only decreased by ~30% on R2. We updated Figure 5C,D and corresponding figure legends to show p-values.

We also measured the level of H3K9me3 on rDNA transgenes and found that it was ~ 4-fold lower compared to endogenous rDNA copies and only slightly above control euchromatin region suggesting that unlike endogenous rDNA, transgenes in euchromatic location do not efficiently acquire heterochromatic marks (Figure 5D, corresponding to Figure 5C in original submission). Furthermore, the levels of heterochromatin mark were similar on intact and damaged rDNA transgenes in control flies, though they expressed at different levels. Finally, upon SUMO KD H3K9me3 levels on intact and R1 transgenes remained unchanged while they are increased on R2 transgene, despite the strong activation of their expression. To summarize, in most of our independent experiments we see either no change or slight increase in H3K9me3 upon SUMO KD, which is opposite of the effect expected if repression would be associated with establishment of H3K9me3 mark. Furthermore, though rDNA transgenes are not enriched in H3K9me3 mark, they still efficiently silenced by SUMO pathway. Finally, there is no difference in H3K9me3 levels between intact and damaged transgenes. Taken together, these results indicate that the level of H3K9me3 mark on rDNA does not correlate with repression.

We completely agree that population measurements using ChIP-seq do not capture the variation between cells: this is equally true for polyploid fly ovaries and any other cell types. Unfortunately, we are not aware of methods that would allow us to reliably profile the levels of H3K9me3 mark on specific loci with single cell resolution. On the other hand, we used sensitive hybridization chain reaction FISH to study expression of rDNA transgenes in individual cells. HCR-FISH showed that depletion of SUMO leads to increased expression of both intact and damaged transgenes in *all* nurse cells (Figure 4B-D). Thus, while ChIP does not provide single cell resolution, analysis of transgene expression indicates that this does not interfere with interpretation of the results: we expect to see decrease in ChIP signal if H3K9me3 mark would be involved in repression. Of note, we are technically able to detect changes in H3K9me3 in nurse cells of ovaries as previously we used ChIP to find a loss of this mark upon disruption of piRNA guided repression (Ninova et al., 2019a, b). We also agree that we are not able to discriminate between all individual rDNA units in the same cell, however, we can reliably discriminate units with R1 and R2 insertions from intact units. As we observed dramatic activation of R1 and R2 upon SUMO KD in all cells we expect to see at least some changes in H3K9me3 on R1 and R2 sequences if this mark is responsible for the repression.

Finally, we agree that results described in original manuscript do not pinpoint specific targets of SUMOylation (proteins and specific residues) that are involved in rDNA repression. We used multi-prong approach, described in detail in our answer to reviewer #1, to get closer to answering this question. Briefly, we showed that recruitment of E2 SUMO ligase Ubc9 to rDNA transgene lead to its repression. Second, we used IP/ Western and mass-spec analyses to identify proteins that are both SUMOylated and involved in nucleolar function and rDNA transcription. Then we tested the roles of identified proteins in rDNA repression using RNAi screen. We found that at least two components of RNA pol I machinery are SUMOylated and are involved in repression. Combined our new results suggest that SUMOylation of potentially several proteins present on rDNA chromatin is responsible for rDNA repression. In the future we plan to further dissect the role of SUMOylation in individual sites of target proteins.

2) It is hard to conclude that repression of rDNA transgenes with insertions differs from cell-to-cell based on their analysis on the highly polyploid nurse cells in the fly ovary (Figure 2C-E), because the analysis cannot distinguish between variations arising from endoreplication vs. differential transcription (same holds for point 1, conclusions about K9me). To rule out this possibility, and to demonstrate conservation of mechanism in non-polyploid tissues, the same analysis should be performed in diploid somatic cells such as neuroblasts. This concern also holds for differences in pre-rRNA expression reported in Figure 4F upon Smt3 knockdown.

Cell-to-cell variability in expression of rDNA transgenes (both intact and with insertions) is a conclusive result of FISH experiments (Figure 2C-E), however, we agree with the reviewer that interpretation of this variability is less straightforward because of polyploidy of nurse cell DNA. We attempted to perform FISH to detect expression of rDNA transgenes in neuroblasts. Unfortunately, no FISH signal was detected in neuroblasts. We think that variation seen in nurse cells is likely due differential expression in individual cells rather than differences in endoreplication for several reasons described below:

1) We showed that overall mechanism of SUMO-dependent rDNA repression is conserved in two different cell types, nurse cells of fly ovary and cultured somatic S2 cells (Figure 3D,E). For example, differences in pre-rRNA expression upon *smt3* knockdown reported in original Figure 4F (Figure 4E in revised manuscript) showed similar changes in both ovary and S2 cells. In this case, RT-qPCR analysis provides population-level picture which does not reveal cell-to-cell variability.

2) For cell-to-cell variability, we confined our analysis to differences in expression between 15 sister nurse cells in the single egg chamber, thus minimizing variability caused by differences in endoreplication. The *Drosophila* endocycle is driven by the oscillations of Cyclin E/Cdk2 activity (Edgar and Orr-Weaver, 2001; Lilly and Duronio, 2005) and individual nurse cells in the egg chamber are inter-connected through ring channels ensuring exchange of macromolecules. In contrast, comparing expression between single cells in other tissues might require their synchronization to eliminate the effects of differences in cell cycle progression on the expression.

3) We used double-FISH to simultaneously monitor expression of rDNA transgenes and control *vasa* gene. While we observe cell-to-cell variability of rDNA transgene expression, we did not detect significant variability of *vasa* expression *in the same cells* (Figure 2C). During revision, we re-analyzed FISH images and conformed that nascent vasa transcripts are expressed at similar levels in the cells that show differences in rDNA transgene expression. Furthermore, FISH with the probes against endogenous prerRNA (many copies combined) show similar levels of expression between individual cells in one egg chamber (Figure 2C). Finally, in our previous studies we used FISH to detect expression of several piRNA transcripts in nurse cells and also did not see cell-to-cell variability. Therefore, cell-to-cell differences in expression seems to be special property of single rDNA transgenes.

4) rDNA transgenes with insertions seem to be completely silent in the majority of cells: on average only 3 out 15 cells in egg chamber have FISH signal while intact transgene *inserted in the same genomic position* is expressed in 10 out of 15 cells (Figure 2D). In contrast, 100% of cells express control vasa gene. Differences in endoreplication are not expected to lead to complete lack of signal in majority of cells. On the other hand, endoreplication should not produce differences between transgenes in identical genomic loci as observed for intact and damaged transgenes.

In revised manuscript we expanded discussion of cell-to-cell variability in rDNA transgene expression to clarify our results and their interpretation.

3) The authors cannot conclude that "insertion of any heterologous sequence into rDNA leads to transcriptional repression", since only one (CFP) was tested in both ovaries and carcasses. Insertions with different lengths and compositions are needed to make this conclusion. More appropriately, the results indicate that integration of a sequence other than transposons can induce rDNA repression.

We agree with the reviewer and have changed the sentence to ‘insertions of a non-transposon CFP sequence lead to comparable decrease in expression, indicating that the repressive effect is not dependent on a specific transposon sequence and can be triggered by heterologous sequence.’

4) I would like to see more information about the functionality of UID-tagged rDNA transcripts. Does the addition of the UID to the rDNA transgene alter its ETS processing kinetics, thus potentially affecting transgene expression measurements? Is the UID marked transcript inside the normal larger nucleolus or in an ectopic nucleolus?

FISH results show that UID marked transcript co-localizes with larger nucleolus (marked by the probe against ETS sequence in pre-rRNA) in majority of the cells (Figure 2C) suggesting its functionality. During the revision we analyzed additional images to confirm this conclusion. We also detected enrichment of RNA pol I on UID-marked transgenes (Figure 5A) indicating that they are transcribed by appropriate rDNA machinery. Thus, rDNA transgenes with UID tag seems to be functional at least during initial steps of their expression. We agree that it would be interesting to explore whether UID-tagged rDNA transcripts are functional at later steps, however, we would argue that later steps of rRNA processing are not relevant to our analysis of rDNA repression.

We employed UID-tagged transgenes to compare expression of intact rDNA and units damaged by R1/R2 insertions. In all our experiments all transgenes that we are directly comparing contain identical UID tag. Therefore, important differences observed between intact and damaged transgenes are not caused by the presence of UID. We did not directly compare UID-marked rDNA transgenes and native rDNA without UID.

5) There is inadequate evidence to support the claim that SUMO is required for "selective repression of damaged rDNA and intact surplus units" (final statement of the Abstract). For example, this claim is contradicted by increased expression of the single intact transgene upon SUMO knockdown (Figure 4A), indicating that SUMO does not selectively repress only "surplus" intact genes or 'damaged rDNA'.

We thank the reviewer for this comment that help us to clarify our message: SUMO-dependent repression controls both intact and damaged units, however, it is biased towards damaged rDNA. We changed the Abstract to properly describe our results and conclusions. Specifically, we changed the sentence in question to ‘Our results suggest that the SUMO pathway is responsible for both repression of damaged units and control of intact rDNA expression’.

[Editors' note: further revisions were suggested prior to acceptance, as described below.]

The manuscript has been improved but there are some remaining issues that need to be addressed before acceptance, as outlined below:Reviewer #3 thought that modifications to the text are desirable before acceptance. In particular, there are points raised about interpretation of the data, and some conclusions need to be tempered or clarified. Their detailed comments can be found below.Reviewer #3:The authors' revisions and new experiments improve the manuscript significantly. Even though there are many mechanistic questions left unanswered, it is important that the results show a critical role for SUMOylation in rDNA silencing. How and why SUMO causes silencing is unknown and awaits future studies. However, there are still issues with the text that need to be fixed. In most cases the authors just need to be more careful about their conclusions.Silencing and K9me. The authors do make a clear case for no or minimal changes in overall levels of H3K9me3 upon SUMO KD. I would still be careful about making the conclusion that H3K9me3 has no role in rDNA silencing. Figure 5C reports that SUMO KD results in an 'only decreased by 36.7% on R2' in the endogenous cluster, but such a change is not trivial and could have impact, especially if the distribution across the rDNA cluster is non-random. Yes, the single rDNA transgenes are derepressed up to 1000 fold by SUMO/Ubc9 KD, and show 'only' a 2.5-fold change in rDNA transcript levels when all 3 HMTases are removed (NB: No legend for Figure 5E…and it is important to know if this figure reports expression from endogenous or transgene rDNA, not clear from the manuscript text either). However, expression does increase 2.5x, and the expectation that the magnitude of the change should be similar may not be true, given the potential involvement of multiple proteins and pathways. If K9me is one but not the only component of rDNA silencing (e.g. if SUMOylation of HP1 and other proteins not directly involved in K9 me is required) SUMO loss would be expected to have a much larger effect. Loss of SUMO modifications on multiple proteins is also consistent with the large number of candidates identified by the screen, which could also be acting additively/synergistically: 'However, the vast majority of affected genes changed within 2-10 fold, in stark contrast to dramatic ~1000- and ~300-fold upregulation of R1 and R2 elements upon SUMO depletion (Figure 3B).'Further, there are reasons to suspect that K9me pathways do impact repression of rDNA. E.g. most rDNA is silent and physically located in the heterochromatin outside the nucleolus. Also, the relationship between this mark and silencing is not straightforward…. for coding genes embedded in heterochromatin, higher levels of K9 me are correlated positively with expression. The point here is to be careful about the interpretation and elimination of a role for K9 me pathways.

We agree with the reviewer about interpretation of the role of H3K9me mark. In revised manuscript we changed discussion of our results and dropped the claim that they rule out a possibility that heterochromatin marks play a role in SUMO-dependent repression. We write that ‘our results indicate that H3K9 methylation might contribute to rDNA silencing along with other pathways’.

We added the legend for Figure 5E.

E3 SUMO ligases. Cannot conclude that 'R1/R2 repression.… is independent of Su(var)2-10 SUMO ligase.' There is precedent even in *Drosophila* for additive effects (redundancy) of SUMO E3s, so one has to look at double and triple mutants in this case as well. Thus it is premature to emphasize possible new E3s over redundancy:"None of tested E3 SUMO ligases scored positive suggesting that SUMO dependent rDNA repression requires yet unknown E3 ligase or involves direct transfer of SUMO by E2 ligase Ubc9 that is indispensable for the repression.'

We agree with the reviewer regarding possible redundancy between different SUMO E3 ligases. Accordingly, we eliminated the claim that R1/R2 repression is independent of Su(var)2-10 ligase. In revised manuscript we discussed a possibility of redundancy between tested E3 ligases.

Variation in endoreplication vs. transcription. The 15 nurse cells do not endoreplicate to the same extent, and there could be even more variation in rDNA levels in these cells given that rDNA is already underreplicated relative to the euchromatin. FISH for rDNA (not RNA) would address this issue definitively. Further, and perhaps more importantly, the expression variability is interesting, but likely specific to the transgenes, since they are not tandemly repeated or in the same heterochromatic environment as the endogenous cluster. The relationship between transgene variability and whether this occurs for individual genes in the endogenous cluster should be discussed.

We agree with the reviewer that endoreplication and unnatural genomic environment might pay a role in cell-to-cell variability of transgene expression. In revised manuscript we added discussion of potential role of endoreplication in transgene variability. We also described a possibility that unnatural genomic environment might be responsible for transgenes expression variability. Finally, we discussed the relationship between transgene variability and expression of endogenous rDNA genes.

Finally, though not a major point, there are many remaining references to 'damaged' rDNA that should be removed, and replaced with 'interrupted' or 'inserted', to avoid reader confusion. This includes the phrase 'damaged by transposon insertions'. Damage strongly implies they are harmed…but being interrupted or silenced is only harmful if it causes cellular or organismal defects. rDNA insertions could be part of an important regulatory network that responds to metabolic changes or stress. There is evidence for such regulation, e.g. Templeton, (1980s) showed changes in the proportion of interrupted units during adaptation to changing environmental conditions, such as dessication (probably via differences in endoreplication). Needs to be looked at and validated with modern eyes and techniques, but still suggests that interrupted rDNA genes could play an important regulatory role and should not be considered 'damaged'.

We removed remaining references to ‘damaged’ rDNA throughout the manuscript and replaced them with ‘interrupted’.

(Henderson and Robins, 1982) is not in the bibliography.

The reference is added to the bibliography.